# Measurements of a potential interference with laser-induced fluorescence measurements of ambient OH from the ozonolysis of biogenic alkenes

Pamela Rickly[1] and Philip S. Stevens[1,2]

[1]School of Public and Environmental Affairs, Indiana, University, Bloomington, IN USA
[2]Department of Chemistry, Indiana University, Bloomington, IN USA

*Correspondence to*: Philip S. Stevens (pstevens@indiana.edu)

**Abstract.** Reactions of the hydroxyl radical (OH) play a central role in the chemistry of the atmosphere, and measurements of its concentration can provide a rigorous test of our understanding of atmospheric oxidation. Several recent studies have shown large discrepancies between measured and modeled OH concentrations in forested areas impacted by emissions of biogenic volatile organic compounds (BVOCs), where modeled concentrations were significantly lower than measurements. A potential reason for some of these discrepancies involves interferences associated with the measurement of OH using the Laser-Induced Fluorescence - Fluorescence Assay with Gas Expansion (LIF-FAGE) technique in these environments. In this study, a turbulent flow reactor operating at atmospheric pressure was coupled to a LIF-FAGE cell and the OH signal produced from the ozonolysis of $\alpha$-pinene, $\beta$-pinene, ocimene, isoprene, and 2-methyl-3-buten-2-ol (MBO) was measured. To distinguish between OH produced from the ozonolysis reactions and any OH artefact produced inside the LIF-FAGE cell, an external chemical scrubbing technique was used, allowing for the direct measurement of any interference. An interference under high ozone (between $2\text{-}10 \times 10^{13}$ cm$^{-3}$) and BVOC concentrations (between approximately $0.1 - 40 \times 10^{12}$ cm$^{-3}$) was observed that was not laser generated and was independent of the ozonolysis reaction time. For the ozonolysis of $\alpha$- and $\beta$-pinene, the observed interference accounted for approximately 40% of the total OH signal, while for the ozonolysis of ocimene the observed interference accounted for approximately 70% of the total OH signal. Addition of acetic acid to the reactor eliminated the interference, suggesting that the source of the interference in these experiments involved the decomposition of stabilized Criegee intermediates inside the FAGE detection cell. Extrapolation of these measurements to ambient concentrations suggests that these interferences should be below the detection limit of the instrument.

## 1. Introduction

The hydroxyl radical (OH) plays an important role in the chemistry of the atmosphere. OH initiates the oxidation of volatile organic compounds (VOCs) which in the presence of nitrogen oxides ($NO_x$) can lead to the production of ozone and secondary organic aerosols, the primary components of photochemical smog. Because of its high reactivity, measurements of OH can provide a rigorous test of our understanding of the fast radical chemistry in the atmosphere. However, several field campaigns have identified significant discrepancies between measured and modeled OH concentrations, especially in low $NO_x$ forested environments (Rohrer et al., 2014). For example, Ren et al. (2008) found that OH concentrations were well predicted by models to within their combined estimated uncertainty when mixing ratios of isoprene were less than approximately 500 pptv, but measurements acquired in areas with higher mixing ratios of isoprene showed observed OH concentrations that were 3-5 times larger than model predictions. Similarly, measurements in a northern Michigan forest found daytime OH concentrations approximately three times larger and nighttime concentrations 3-10 times larger than model predictions (Tan et al., 2001; Faloona et al., 2001). Aircraft measurements over the Amazon rainforest found OH concentrations to be 40-80% larger than model predictions (Lelieveld et al., 2008), while measurements of OH in the Borneo rainforest were 5-10 times greater than model predictions (Whalley et al., 2011). Similarly, measurements of OH concentrations under high mixing ratios of isoprene in the Pearl River Delta, China were 3-5 times larger than model predictions (Hofzumahaus et al., 2009).

Most of these measurements were done using the Laser-Induced Fluorescence – Fluorescence Assay by Gas Expansion (LIF-FAGE) technique. In this technique, ambient air is sampled through an inlet at low pressure, enhancing the OH fluorescence lifetime and allowing temporal filtering of the OH fluorescence from laser scatter (Heard and Pilling, 2003). Fluorescence from OH radicals is distinguished from background scatter and broadband fluorescence through spectral modulation of the laser wavelength. Previous laboratory tests using the Penn State instrument demonstrated that the technique was free from interferences from several species, including spectral interferences from naphthalene, sulfur dioxide, and formaldehyde as well as chemical interferences from high concentrations of $H_2O_2$, HONO, $SO_2$, $HNO_3$, several alcohols and alkanes, propene, and isoprene (Ren et al., 2004). Mixtures of ozone with ethene, propylene and isoprene did not result in any significant signal, suggesting that the ozonolysis of these compounds did not produce an interference in the Penn State instrument, although small interferences were observed with addition of high amounts of ozone and acetone that would be insignificant under ambient conditions (Ren et al., 2004). Measurements of OH in the SAPHIR chamber at atmospheric pressure and under varying concentrations of $H_2O$, $O_3$, CO, HCHO, NO, and $NO_2$ by both an LIF-FAGE instrument and a differential optical absorption spectroscopy (DOAS) instrument were in excellent agreement,

suggesting that measurements of OH using the LIF-FAGE instrument were free from artifacts under these conditions (Schlossler et al., 2007). A subsequent intercomparison inside the SAPHIR chamber found that measurements from several different LIF-FAGE instruments under a variety of conditions agreed with each other to within the calibration accuracies of the instruments (Schlosser et al., 2009). No interferences were detected under varying concentrations of $O_3$, $H_2O$, $NO_x$ and peroxy radicals, and measurements of OH during the ozonolysis of various mixing ratios of pent-1-ene (6-25 ppb) and trans-2-butene (200 ppb) in approximately 100 ppb of ozone also did not reveal a significant interference (Schlosser et al., 2009). In contrast, Hard et al. (2002) observed an interference in their LIF-FAGE instrument during calibrations using the ozonolysis of trans-2-butene under high mixing ratios of both ozone and trans-2-butene. Tests suggested that the interference was not laser generated, and disappeared in air containing 1% water vapor. They suggested that the interference may be due to the dissociation of an intermediate in the ozonolysis mechanism that produces OH in the low-pressure cell of their FAGE instrument (Hard et al., 2002).

Recently, Mao et al. (2012) discovered a significant interference associated with their LIF-FAGE measurements of OH in a northern California forest. Using a chemical scavenger to remove ambient OH before air enters the inlet, they found that subsequent spectral modulation revealed a significant amount of internally generated OH from an unknown interference. Measurements using only spectral modulation of the laser wavelength were greater than the measurements when the interference measured using the chemical scavenger was subtracted, and the latter measurements were in better agreement with model predictions (Mao et al., 2012). Similar results were observed by Novelli et al. (2014a) who measured ambient OH concentrations in several forest environments using an external chemical scavenger technique. They found that OH generated inside their detection cell comprised 30-80% of the daytime signal observed using spectral modulation and 60-100% of the signal observed at night. Subtracting the interference from the ambient signal resulted in OH concentrations that were in good agreement with model simulations as well as measurements by a Chemical Ionization Mass Spectrometry (CIMS) instrument (Novelli et al., 2014a).

Mao et al. (2012) found that the interference increased with both temperature and measured OH reactivity, suggesting that the interference was related to biogenic emissions in this environment, perhaps the result of BVOC oxidation products entering the sampling cell of the LIF-FAGE instrument which subsequently undergo further reactions and/or decomposition producing the additional OH signal. Novelli et al. (2016) also found that the interference appeared to correlate with temperature similar to the temperature dependence of terpene emissions and was also correlated with the measured ozone concentrations. They concluded that one possible contributor to their interference was the decomposition of stabilized Criegee intermediates inside their instrument.

In addition to decomposition leading to OH formation at ambient pressure (Fang et al., 2016; Kidwell et al., 2016; Smith et al., 2017), previous laboratory experiments have shown that BVOC ozonolysis intermediates, such as Criegee intermediates and vinyl hydroperoxides, are likely to promptly decompose to produce OH at low pressures (Kroll et al., 2001a,b). As a result of the large pressure and temperature gradients which occur as the sample enters the FAGE detection cell, the dissociation pathway of these intermediate species may be favored, leading to additional OH production. Recently, Fuchs et al. (2016) performed laboratory and chamber experiments to determine whether the ozonolysis of alkenes produced an OH artifact in their LIF-FAGE instrument. They found that under reactant concentrations that were orders of magnitude greater than ambient, the ozonolysis of propene, α-pinene, limonene, and isoprene produced a detectable interference in their instrument that increased with the turnover rate of the reaction. Extrapolating their results to ambient concentrations of ozone and alkenes would suggest that the ozonolysis of these compounds would not produce a detectable interference in their instrument under ambient conditions (Fuchs et al., 2016).

The goal of this work is to determine whether intermediates or products in the ozonolysis of various biogenic alkenes can lead to an interference with OH measurements using the Indiana University LIF-FAGE instrument (IU-FAGE). These experiments focus on the ozonolysis of several biogenic alkenes, including α-pinene, β-pinene, ocimene, isoprene, and 2-methyl-3-buten-2-ol (MBO), all of which have been observed to contribute appreciably to BVOC emissions in forested environments (Guenther et al., 1994; Harley et al., 1998). Measurements of the interference as a function of various instrumental parameters are also provided in an attempt to identify possible sources of the interference and ways it could be minimized.

## 2. Experimental Section

The ozonolysis experiments were performed using an atmospheric pressure turbulent flow tube with a movable injector similar to that used for ambient measurements of total OH reactivity (Hansen et al., 2014). The 1 m long and 5 cm diameter flow tube was positioned perpendicular to the IU-FAGE detection cell so that the flow would not interfere with the external OH scavenging measurement (section 2.2) (Fig. 1). A Teflon adaptor attached at one end of the flow tube supported the injector, a 1 m stainless steel tube with a 1.25 cm diameter. This injector allowed for the introduction of ozone produced from an ozone generator (Enaly) to the system and could be moved throughout the flow tube to permit varying reaction times between approximately 100 and 420 ms calculated based on the measured velocity in the flow tube. Attached to the end of the injector was a turbulizer used to increase mixing of the reagents at the start of the reaction. A flow of nitrogen of 180 SLPM created a turbulent

flow with a Reynolds number of approximately 3750. Ozone concentrations were varied between approximately 1 and 3 ppm ($2\times10^{13}$ - $7\times10^{13}$ molecules cm$^{-3}$), with most measurements occurring at ozone concentrations of 2.5, 5, and $7\times10^{13}$ molecules cm$^{-3}$ as the ozone generator provided the highest stability at these concentrations. Ozone concentrations in the flow tube were measured using a Teledyne Photometric Ozone Analyzer (model 400E), with an estimated uncertainty of $\pm0.5$ ppb.

BVOC concentrations were introduced into the flow tube by bubbling $N_2$ through the liquid compound sending the vapor into the reactor. The concentration of the BVOC was estimated from its equilibrium vapor pressure and accounting for dilution into the main flow. Several alkene concentrations were used for each experiment, with approximate concentrations of $2\times10^{11}$ to $4\times10^{13}$ molecules cm$^{-3}$ for $\alpha$-pinene (Aldrich, 98%), $1\times10^{11}$ to $4\times10^{13}$ molecules cm$^{-3}$ for $\beta$-pinene (Aldrich, 99%), $9\times10^{10}$ to $5\times10^{13}$ molecules cm$^{-3}$ for ocimene (Aldrich, mixture of isomers, $\geq90\%$), and approximately $8\times10^{13}$ molecules cm$^{-3}$ for isoprene (Aldrich, 99%), and MBO (Aldrich, 98%). Unfortunately, no direct method for measuring the concentration of these BVOCs was available, and as a result the absolute concentration of BVOCs in the flow tube is highly uncertain due to potential wall losses prior to entering the flow tube.

## 2.1 Detection of OH Radicals

OH radicals were measured using the IU-FAGE instrument, in which ambient air is pulled through either a 0.6 or 1 mm diameter nozzle and expanded to a total pressure of approximately 4-9 Torr resulting in a total flow rate of approximately 8-10 SLPM (Dusanter et al., 2008; 2009; Griffith et al., 2013; 2016). Previous field measurements using the IU-FAGE instrument have incorporated a cylindrical inlet (5 cm diameter, 14 cm long) attached to the main detection block resulting in a total distance of approximately 20 cm from the nozzle to the detection volume (Fig. 2). These previous field measurements have utilized both the 0.6 mm nozzle (Griffith et al., 2016) and the 1 mm nozzle (Dusanter et al., 2009; Griffth et al., 2013), and the experiments presented here have attempted to reproduce these instrument configurations.

The original IU-FAGE laser system used in this study consisted of a Spectra Physics Navigator II YHP40-532Q diode-pumped Nd:YAG laser that produces approximately 5.5W of radiation at 532 nm at a repetition rate of 5 kHz. This laser pumped a Lambda Physik Scanmate 1 dye laser (Rhodamine 640, 0.25 g L$^{-1}$ in isopropanol) that produced tunable radiation around 616 nm, which was frequency doubled to generate 2 to 20 mW of radiation at 308 nm. This laser system was recently replaced with a Spectra Physics Navigator II YHP40-532Q that produces approximately 8 W of radiation at 532 nm at a repetition rate of 10 kHz that pumps a Sirah

Credo Dye laser (255 mg/L of Rhodamine 610 and 80 mg/L of Rhodamine 101 in ethanol), resulting in 40 to 100
mW of radiation at 308 nm.  Initial measurements of the OH interference from ocimene and α-pinene ozonolysis
were made using the original laser system, with the majority of experiments done using the newer laser system.

4         After exiting the dye laser, the beam is focused onto a 12 m optical fiber to transmit the radiation to the

sampling cell where it crosses the expanded air perpendicular to the flow approximately 24 times in a multipass
White cell configuration (Fig. 2). The OH molecule is excited and detected using the $A^2\Sigma^+$ (v´= 0)←$X^2\Pi_i$ (v´´=
0) transition near 308 nm. A reference cell where OH is produced by thermal dissociation of water vapor is used
to ensure that the laser is tuned on-line and off-line of the OH transition to measure the net fluorescence signal.
The OH fluorescence is detected by a gated microchannel plate detector (Hamamatsu R5916U-52) and the
resulting signal is sent through a preamplifier (Stanford Research SR445) and a photon counter (Stanford Research
SRS 400). The detector is switched off during the laser pulse through the use of electronic gating, allowing the
OH fluorescence to be temporally filtered from laser scatter. Each offline measurement is recorded for
approximately 10 seconds and is averaged and subtracted from the online measurement, averaged for
approximately 20 seconds. These measurements are recorded for at least 5 cycles per ozone concentration once
the OH concentration has stabilized after several minutes.

16        The sensitivity of the IU LIF-FAGE instrument was calibrated using the UV-water photolysis technique

where water vapor is photolyzed to produce known amounts of OH (Dusanter et al., 2008). In these experiments,
calibrations were performed under the conditions of the experiments using $N_2$, resulting in larger calibration
factors compared to ambient air due to fluorescence quenching by oxygen. Nitrogen from liquid boil-off was used
instead of synthetic air to reduce the concentration of reactive impurities in the flow tube. The $N_2$ calibration
factors were determined for various instrumental parameters including three cell pressures (4, 7, and 9 Torr), the
two nozzle diameters (0.6 mm and 1 mm), and three inlet lengths (3.2 cm, 14 cm, and 24.8 cm) (Table S1, Fig.
S1). For reference, previous ambient measurements using the IU LIF-FAGE instrument were typically acquired
using the medium inlet length (Fig. 2) and both the 0.6 and 1 mm nozzle diameters, resulting in cell pressures of
4-7 Torr depending on the nozzle diameter size and pumping efficiency (Lew et al., 2017a).  The error associated
with the UV-water photolysis calibration technique is estimated to be ±36% (2σ) (Dusanter et al., 2008)

27        Both nozzle diameters showed similar sensitivities that decreased as the pressure increased due to

increased collisional quenching of the OH fluorescence. The calibration factor was also sensitive to the length of
the inlet where the ambient air enters the cell and where the OH fluorescence occurs in the detection axis, with
the sensitivity decreasing with the increasing inlet length likely due to increased loss of OH radicals on the interior
walls of the inlet. For these experiments, the limit of detection was between approximately $5\times10^5$ - $4\times10^6$
molecules cm$^{-3}$ (S/N=1, 10 min integration) depending on the inlet configuration, flow rate, and pressure inside
the FAGE detection cell, with the lowest value corresponding to the shortest inlet and lowest pressure, and the
highest value corresponding to the longest inlet and highest pressure.
**2.2 Measuring the OH Interference**
The OH interference was measured using a chemical titration scheme in which perfluoropropylene ($C_3F_6$, 99.5%,
Matheson) was added through a circular ring surrounding the detection inlet to chemically remove external OH
(Griffith et al., 2016) (Fig. 2). Measurements of OH concentrations using spectral modulation compared to
chemical modulation with $C_3F_6$ added externally can reveal OH radicals that are generated inside the detection
cell. $C_3F_6$ was used because it reacts quickly with OH while also having a negligible optical absorption around
308 nm (Mao et al. 2012). The injector is approximately 3.5 cm in diameter and 1.0 cm in height and consists of
10 equally spaced 0.4 mm holes in the ring that surround the inlet nozzle at the center of the injector.  The holes
are approximately 0.5 cm in above the level of the detection cell inlet. Increasing the height of the injector above
the inlet from 0.5 cm to 4.5 does not impact the overall sensitivity of the instrument, suggesting that OH radicals
are not lost on the outer walls of the injector.
To determine the flow of $C_3F_6$ to be used, OH was produced from the photolysis of ambient air using a
mercury penlamp placed in front of the inlet (Fig. S2). Once a constant OH signal was established, $C_3F_6$ was added
at varying flows to determine the flow that depleted ≥90% of the external OH signal (a total flow of 99.5% $C_3F_6$
of approximately 3-5 sccm). With the flow entering the detection cell inlet of 3-9 SLPM depending on inlet
diameter size, this flow of $C_3F_6$ results in an approximate concentration of $1\text{-}3 \times 10^{16}$ molecules cm$^{-3}$ and a
residence time of approximately 0.05 s. This flow rate was able to titrate the externally generated OH regardless
of the flow rate set by each individual inlet diameter. To ensure that this flow of $C_3F_6$ did not titrate OH radicals
produced inside the detection cell, a penlamp was placed inside the detection cell directly behind the inlet to
generate OH radicals internally. The same external $C_3F_6$ flow was again introduced to ensure that the concentration
of $C_3F_6$ after expansion into the detection cell was not high enough to titrate any internally generated OH.
By applying this $C_3F_6$ titration method to the alkene ozonolysis experiments, any OH produced in the
flow tube was expected to be removed. Although the ozonolysis reactions may still lead to additional OH
production during the residence time in the injector, kinetic simulations suggest that under the conditions of these
experiments the steady-state concentration of OH in the presence of $C_3F_6$ would be near or below the detection
limit of the instrument (less than $5 \times 10^5$ molecules cm$^{-3}$). However, for some of the high concentration
experiments, the amount of $C_3F_6$ added may not have been sufficient to reduce the steady-state concentration of
OH to below the detection limit of the instrument, especially for the ocimene experiments due to the high reactivity
of ocimene with ozone. However, the model simulations suggest that even for these high concentration
experiments the remaining steady-state OH concentrations represented less than 10% of the observed interference.
Thus, the majority of OH that was measured using spectral modulation after $C_3F_6$ addition would be an
interference generated internally. Subtraction of this interference from the measurement acquired before $C_3F_6$
addition should reflect the steady-state OH concentration generated in the flow tube, which can then be compared
to literature values of the OH yield for the ozonolysis of these alkenes.
**3. Results and Discussion**
**3.1 Ozonolysis experiments**
The interference tests were performed during the alkene ozonolysis experiments with a reaction time in the flow
tube of approximately 420 ms, a reaction time longer than that required for the system to reach steady-state (less
than approximately 20 ms based on model calculations), and the results are shown in Figs. 3 and 4. In these
figures, the open symbols are the measured OH concentration produced from the ozonolysis reaction without
addition of $C_3F_6$ (With interference), and the filled symbols represent the OH signal after the signal measured with
$C_3F_6$ addition is removed (Without interference) using the 0.6 mm nozzle, shown on the same plot to illustrate the
magnitude of the interference. The expected steady-state OH concentrations based on previous measurements of
the OH yield for each compound are also shown in each figure by the solid line, calculated using the following
equation:

$$[OH]_{ss} = \frac{k_{O_3+VOC}\alpha[O_3][alkene]}{k_{OH+VOC}[alkene] + k_{OH+O_3}[O_3] + k_{wall}} \approx \frac{k_{O_3+VOC}\alpha[O_3]}{k_{OH+VOC}} \tag{1}$$

In this equation, $k_{O3+VOC}$ is the rate constant for the $O_3$ + alkene reaction with an OH yield of $\alpha$, $k_{OH+VOC}$ is the rate
constant for the OH + alkene reaction, $k_{OH+O3}$ is the rate constant for the OH + $O_3$ reaction, and $k_{wall}$ is the first-
order loss of OH on the walls of the reactor, measured as described in Hansen et al. (2014). Determining the
expected steady-state OH concentration from this equation has the advantage of being independent of the
concentration of each BVOC. Rate constants for the OH and $O_3$ reactions were obtained from recommendations
by Atkinson (1997) and Atkinson et al. (2006). For these calculations, the loss of OH from reaction with ozone
(approximately 7 s$^{-1}$) as well as wall loss of OH (approximately 3.6 s$^{-1}$), were neglected, as they were much smaller
than loss of OH due to reaction with the alkenes due to the high concentration of alkenes used in these experiments.
As illustrated in Fig. 3, the measured OH concentration from the ozonolysis of α-pinene without $C_3F_6$
addition is consistently greater than the measurements after the interference is subtracted, indicating that a
significant concentration of OH is being produced inside the detection cell. In these figures, instabilities in the
concentration of ozone produced by the ozone generator resulted in small variations in the measured ozone
concentrations between measurements with and without added $C_3F_6$. Experiments without α-pinene in which
ozone alone was sampled by the detection cell showed a negligible OH signal, suggesting that the interference
was related to the presence of both the alkene and ozone. The observed interference measured with the addition
of $C_3F_6$ accounted for approximately 40-60% of the observed signal in the absence of $C_3F_6$. When the measured
interference was subtracted from the overall signal, the resulting OH concentrations were in reasonable agreement
with the OH yield expected from α-pinene ozonolysis resulting in an OH yield of approximately $0.81 \pm 0.10$,
determined from a weighted fit of the slope of the plot of the OH concentration versus ozone concentration, in
good agreement with the value of $0.76 \pm 0.11$ as reported by Chew and Atkinson (1996) and the value of $0.91 \pm$
$0.23$ reported by Siese et al. (2001).

14       The measured interference appeared to depend on the length of the inlet, with the greatest interference

observed with the longest inlet (Fig. 3), while the interference measured with the short and medium length inlets
were comparable. Similar results were observed with the use of the 1 mm nozzle diameter (Fig. S3), with the
longest inlet displaying the greatest interference. These results are similar to that observed by Fuchs et al. (2016),
who found that the interference from the ozonolysis of α-pinene, limonene and isoprene also increased with the
length of the inlet in their FAGE instrument. But while the inlet length appears to exhibit a trend in the OH
interference, the cell pressure did not appear to significantly impact the level of the interference except at the
longest inlet length. These results are in contrast to the results of Fuchs et al. (2016), who found that the
interference in their instrument decreased with cell pressure, although the effect was greater with the longer inlet.
The different cell pressures in these experiments were obtained by changing the pumping speed, which impacts
the velocity and residence time of the airstream inside the detection cell. This suggests that increasing the reaction
time for the short and medium length inlets does not significantly impact the interference, while increasing the
reaction time for the longest inlet does increase the interference. However, at the longest inlet length, the increased
reaction time likely also leads to increased collisions with the interior surfaces of the detection cell, which could
also lead to increased dissociation and production of OH. Increasing the nozzle diameter decreases the flow
velocity inside the detection cell, which increases the reaction time and the frequency of wall collisions using the
longer inlet length. Although this also increases the loss of OH generated by the interference, it appears that the
rate of production of OH by the interference is greater than the loss rate of OH on the walls of the inlet. These
results appear to be independent of the α-pinene concentration (Fig. 4, top) as the level of the interference is
similar for estimated α-pinene concentrations between approximately 1 and $4\times10^{13}$ molecules cm$^{-3}$.
The results from the ozonolysis of β-pinene are also shown in Fig. 4 (middle). Similar to that observed
for the ozonolysis of α-pinene, the observed interference measured with the addition of $C_3F_6$ accounted for
approximately 40% of the observed signal in the absence of $C_3F_6$. When the measured interference was subtracted
from the overall signal, the measured OH concentrations results in an overall OH yield of approximately 0.40 ±
0.01, which is in good agreement with the value reported by Atkinson et al. (1992) of 0.35 ± 0.05. The percentage
of interference as well as the resulting OH yield was relatively constant for each β-pinene concentration used;
however, the results suggest that the OH yield may increase with increasing concentrations of β-pinene.
Measurements of the concentration of OH produced from the ozonolysis of various concentrations of
ocimene are also shown in Fig. 4 (bottom). In contrast to the ozonolysis of α-pinene, the interference and the
observed OH yield appear to increase with increasing ocimene concentration. At ocimene concentrations of
approximately $4\times10^{11}$ molecules cm$^{-3}$, an OH yield of approximately 0.28 ± 0.02 was measured. However, the
apparent OH yield increased to a yield of 1.0 ± 0.03 at the highest ocimene concentration (approximately $5\times10^{13}$
molecules cm$^{-3}$). Additional ocimene ozonolysis experiments were performed using the two nozzle diameters and
three inlet lengths with the results shown in Fig. S4. These results show that the measured OH yield from ocimene
ozonolysis after subtracting the interference is only consistent with the results of Aschmann et al. (2002) of 0.55
± 0.09 for an ocimene concentration of approximately $1\times10^{13}$ molecules cm$^{-3}$. The increase in the apparent OH
yield and interference as the concentration of ocimene was increased may be due to the additional ozonolysis of
the reaction products, as the products of ocimene ozonolysis are unsaturated. The higher ocimene concentrations
would lead to increased concentrations of reaction products, which in turn could also contribute to the measured
interference and OH yield. Additional experiments are needed in order to better quantify the OH yield from the
ozonolysis of ocimene.
In contrast to the measurements involving α-pinene, β-pinene, and ocimene, similar ozonolysis
experiments involving isoprene and MBO did not produce a detectable OH signal or interference under the highest
ozone and alkene concentrations. These results may reflect the lower relative reactivity of isoprene and MBO with
$O_3$ compared to reaction with OH as well as a lower yield of OH under the conditions of these experiments.
Simulations of the kinetics using the MCM mechanism for the ozonolysis of isoprene result in steady-state OH
concentrations that are a factor of 50 lower than that predicted for the ozonolysis of α-pinene, consistent with a
lower steady-state OH concentration estimated using Equation 1, and are near or below the detection limit of the
instrument. However, these results do not rule out the possibility of an interference from the ozonolysis of these
compounds. The lower reactivity of isoprene and MBO with ozone would likely require longer reaction times or
higher concentrations of ozone and/or alkenes to establish steady-state OH concentrations above the detection
limit of the instrument in these experiments. Fuchs et al. (2016) did find a significant unexpected OH signal in
laboratory under very high concentrations of isoprene and a reaction time on the order of 1 s. Future experiments
will involve higher concentrations of ozone, isoprene and MBO and/or longer reaction times to increase the
ozonolysis rate for these reactions and to determine the potential magnitude of the interference from isoprene and
MBO in the IU-FAGE instrument.
**3.2 Interference measurements as a function of laser power, ozone, and ozonolysis reaction time**
A potential interference with the detection of OH radicals by laser-induced fluorescence is the production of OH
through photolysis of ozone by the laser, with the resulting excited oxygen atoms reacting with ambient water
vapor to produce OH (reactions R1 and R2) (Stevens et al., 1994; Ren et al., 2004):

$$O_3 + hv \rightarrow O(^1D) + O_2 \tag{R1}$$

$$O(^1D) + H_2O \rightarrow OH + OH \tag{R2}$$

The resulting OH signal from this two-photon process would display a quadratic dependence on laser power. One
possible source of the interference observed in the ozonolysis experiments could be $O(^1D)$ produced from the
photolysis of ozone by the laser reacting with the alkene leading to the production of OH radicals through a
hydrogen abstraction mechanism. The OH signal from this two-photon process would also display a quadratic
dependence on laser power.
To confirm that laser generated OH was not occurring within this instrument and was not the source of
the interference, experiments were conducted over a wide range of laser powers (0.6-12 mW) to determine
whether the observed interference displayed a dependence on laser power consistent with a laser-generated
mechanism. Fig. 5 (top) shows the measurements of the OH concentration with and without the interference for
the ozonolysis of ocimene as a function of laser power. These experiments were done under dry conditions to
minimize potential laser-generated interferences from reactions R1 and R2. Because the OH concentration
remained relatively stable over the range of laser powers used in these experiments, there is no indication that the
interference was due to a laser-generated mechanism. These results are similar to that found by Novelli et al.
(2014a) and Fuchs et al. (2016).
A plot of the interference at different ozone concentrations is also shown in Fig. 5 (middle) for the
ozonolysis of α-pinene as an example. Comparing the measured interference for each inlet length and ozone

concentration shows a trend in which the interference appears to increase with increasing ozone concentration, with the greatest increase occurring with the longest inlet. Similar to the results of Fuchs et al. (2016), the observed interference appears to increase with the overall ozonolysis turnover time, defined as the steady-state rate of OH radical propagation, expressed here as the rate of alkene ozonolysis (Fig. 6). Although the magnitude of the interference observed here is greater due to the greater turnover rates used in these experiments, the slope of the observed interference using the short inlet as a function of the turnover rate appears to be similar to that observed by Fuchs et al. (2016), although the results presented here are highly scattered, likely due to the large uncertainty associated with estimating the concentration of each BVOC. However, the level of the observed interference is greater than that illustrated in Fig. 6 for the measurements using the long inlet, suggesting the similarity with the results of Fuchs et al. (2016) may be fortuitous, as differences in the design of the instrument impacts the level of the interference. In contrast to the results of Fuchs et al. (2016), the level of interference as a function of the turnover rate is not similar for all of the BVOCs tested. While the observed interference as a function of turnover rate appears to be similar for the ozonolysis of α-pinene and ocimene, the observed interference for the ozonolysis of β-pinene is significantly less. This may also be related to uncertainties associated with estimates of the concentration of β-pinene in the flow tube, but may also suggest differences in the mechanism for the production of the interference for some BVOCs. Additional measurements of the interference for other BVOCs are needed to resolve this discrepancy.

The interference signals from the ozonolysis of α-pinene, β-pinene, and ocimene for each experimental condition are summarized in Table S2. The interference from α-pinene expressed as a percentage of the total OH signal at several ozone concentrations and cell pressures for the short, medium, and long inlets are shown in Fig. S5. On average, the percent interference was similar for ozone mixing ratios between 1-3 ppm, with values of approximately $45 \pm 3\%$, $37 \pm 8\%$, and $58 \pm 8\%$ for the short, medium, and long inlets, respectively for the 1 mm nozzle, and approximately $54 \pm 7\%$, $50 \pm 13\%$, and $65 \pm 11\%$ for the short, medium, and long inlets, respectively for the 0.6 mm nozzle. As discussed above, the similarity of the measured interference between the short and medium length inlets suggest that the level of interference is not directly related to the residence time inside the FAGE detection cell, but may be the result of increased collisions with the interior surfaces of the detection cell that occurs when using the longest inlet. Similar results were observed for the ozonolysis of ocimene (Fig. S6), although additional experiments will be needed to determine whether a similar trend with inlet length and residence time is observed.

The interference was also measured over varying reaction times within the flow tube, and an example of the results is shown in Fig. 5 (bottom). The results appear to indicate that the level of interference does not depend

on the ozonolysis reaction time in the flow tube. If the interference was due to a stable oxidation product, it would
be expected to increase with reaction time as the concentration stable oxidation products accumulated in the flow
tube. These results suggest that the interference is not due to a stable oxidation product but may instead be due to
a steady-state intermediate in the ozonolysis mechanism. The consistency of the measured interference relative to
the OH concentration produced from the ozonolysis mechanism may suggest that the source of the interference is
related to the source of OH, such as the Criegee intermediate.
**3.3 Stabilized Criegee intermediates as a source of the interference**
*3.3.1 Decomposition of Criegee intermediates produced inside the FAGE detection cell as a source of*
*the interference.* The ozonolysis of alkenes involves the addition of ozone to the double bond, forming a primary
ozonide which quickly decomposes to an excited Criegee intermediate and a carbonyl compound. Depending on
its structure, the excited Criegee intermediate can either decompose to form an OH radical, or can be stabilized
by collisions forming a stabilized Criegee intermediate (SCI), which can also thermally dissociate into OH at
longer reaction times (approximately 1 sec for tetramethylethylene, Kroll et al., 2001b). At shorter reaction times,
the excited Criegee intermediate is in steady-state with respect to dissociation and stabilization, resulting in OH
concentrations that are generally independent of reaction time, but increase at longer reaction times due to the
additional OH production from SCIs (Kroll et al., 2001b). Rate constants for decomposition and stabilization of
excited Criegee intermediates are estimated to be on the order of $10^4$ -$10^9$ s$^{-1}$ (Horie and Moortgat, 1991), while
the rate constant for decomposition of stabilized Criegee intermediates have been measured to be on the order of
$10^2$ s$^{-1}$ (Smith et al., 2016). As a result, the concentration of excited Criegee intermediates are expected to be very
low compared to the concentration of SCIs, which have estimated lifetimes on the order of milliseconds relative
to decomposition. The atmospheric fate of SCIs will depend on the rate of decomposition relative to reaction with
other atmospheric trace gases such as $SO_2$ or $H_2O$ among others (Novelli et al., 2014b). This lifetime may be
longer in the low pressure region of the FAGE detection cell due to the decrease in concentration of reactants such
as $SO_2$ or $H_2O$, allowing these SCIs to collide with the walls of the detection cell.

25           The time dependence of the observed interference in these experiments, with the interference appearing

to be independent of reaction time except under the longest reaction time, at first appears be consistent with a
mechanism that involves the formation of excited Criegee radicals inside the FAGE detection cell. However, it is
unlikely that excited Criegee intermediates could be produced directly from alkene ozonolysis inside the IU-
FAGE detection cell as the decrease in reactant concentrations from the expansion to low pressure leads to
turnover rates that are approximately two orders of magnitude smaller than that shown in Fig. 6. For the ozonolysis

of α-pinene, a reaction time of approximately 0.5-1 s would be required to internally produce the observed OH signals in these experiments, which is much longer than the reaction time inside the detection cell (on the order of 1-2 ms). The short reaction time in the detection cell is also too short for OH concentrations to reach steady-state. The lower pressure in the FAGE detection cell may result in higher yields of OH due to the lower rate of stabilization of the excited Criegee intermediate under these conditions (Kroll et al., 2001a). Nevertheless, an OH yield of one would still require a reaction time of approximately 0.18 to 0.35 s to produce the observed internal OH concentrations.

*3.3.2 Decomposition of Criegee intermediates produced outside of the FAGE detection cell as a source of the interference.* Previous measurements have demonstrated that the stabilized Criegee intermediates produced external to the FAGE detection cell from the ozonolysis of propene and (*E*)-2-butene can decompose to produce OH radicals upon entering the low pressure region of the FAGE detection cell (Novelli et al., 2014b). The time dependence of the OH production was consistent with previous measurements of the rate of unimolecular decomposition of SCIs (Novelli et al., 2014b). To determine whether the decomposition of stabilized Criegee intermediates are the source of the interference in these experiments, acetic acid was added to the flow tube as an SCI scavenger. Welz et al. (2014) have reported direct measurements of the rate constants for the reactions of $CH_2OO$ and $CH_3CHOO$ Criegee intermediates with formic and acetic acids and found values in excess of $1 \times 10^{-10}$ $cm^3$ molecule$^{-1}$ s$^{-1}$. In contrast, the rate constant for reaction of acetic acid with OH is approximately $7 \times 10^{-13}$ $cm^3$ molecule$^{-1}$ s$^{-1}$ (Atkinson et al., 2006). Acetic acid was chosen over $SO_2$ or water as an SCI scavenger as $SO_2$ fluorescence at 308 nm can interfere with OH measurements (Fuchs et al., 2016), while the addition of water in the presence of the high concentrations of ozone in these experiments would have led to significant laser-generated OH from reactions R1 and R2. Assuming that other Criegee intermediates react similarly, addition of acetic acid should scavenge SCIs in the flow tube. If SCIs are the source of the OH interference, the measured OH signal resulting from addition of acetic acid should be equivalent to the OH signal when the OH interference measured using external $C_3F_6$ titration is subtracted. For these experiments, approximately $9 \times 10^{12}$ molecules $cm^3$ of acetic acid was introduced into the flow tube, and allowed to react for approximately 200 ms. At this concentration, the reaction with acetic acid was modeled to have a minimal impact on the steady-state concentration of OH.

The results of these experiments for the ozonolysis of α-pinene and ocimene are shown in Fig. 7. In this figure, the open symbols are the measured total OH signal due to OH produced in the flow tube from both excited and stabilized Criegee intermediates plus the OH interference, while the solid red symbols represent the remaining OH signal after the interference (measured after external $C_3F_6$ addition) is subtracted. The solid green symbols represent the measured OH signal (without external $C_3F_6$ addition) when acetic acid is added to the flow tube. As

can be seen from this figure, the measured OH signal when acetic acid is added is similar to the OH signal when the interference is subtracted, suggesting that the interference is due to the decomposition of SCIs inside the IU-FAGE detection cell, consistent with the results of Novelli et al. (2014b). These results also suggest that the majority of the OH radicals in the flow tube are produced from the rapid decomposition of excited Criegee radicals. If stabilized Criegee intermediates were responsible for a significant fraction of the OH produced in the flow tube, it is likely that addition of acetic acid would have led to measured OH signals that were significantly lower than the OH signal after the interference was subtracted. However, these results do not exclude the possibility that SCIs are also thermally decomposing and contributing to OH production in the flow tube. Recent measurements suggest that the unimolecular decomposition of the $(CH_3)_2COO$ Criegee intermediate can be rapid and compete with reaction of this intermediate with water or $SO_2$ (Smith et al., 2016; Chhantyal-Pun et al., 2017). A similar rate for the decomposition of SCIs in the experiments reported here could also compete with reaction of this intermediate with acetic acid. Additional experiments are needed to determine whether stabilized Criegee intermediates contribute to OH radical production in the ozonolysis mechanisms of these biogenic compounds.

To determine whether the magnitude of the interference observed in these experiments was consistent with the concentration of stabilized Criegee intermediates in the flow tube, the concentration of these intermediates was estimated from a chemical model of the ozonolysis of α-pinene using the Master Chemical Mechanism (MCM v3.2, Jenkin et al., 1997; Saunders et al., 2003). In this mechanism, the ozonlysis of α-pinene results in the formation of two Criegee intermediates, APINOOA and APINOOB. APINOOA decomposes to OH radicals through two channels with different co-products with a decomposition rate constant of $1 \times 10^6$ s$^{-1}$. APINOOB also decomposes to form an OH radical with a decomposition rate constant of $0.5 \times 10^6$ s$^{-1}$, but can also form a stabilized Criegee intermediate (APINBOO) with a rate constant of $0.5 \times 10^6$ s$^{-1}$. In addition to reaction with CO, NO, $NO_2$, and $SO_2$ (which are not significant in these experiments), the stabilized Criegee intermediate in this mechanism can also decompose to form pinonaldehyde or pinonic acid, with rate constants that depend on the concentration of water vapor ($1.40 \times 10^{-17}*H_2O$ and $2.00 \times 10^{-18}*H_2O$ s$^{-1}$, respectively). This version of the MCM does not include a mechanism for the formation of OH from the stabilized Criegee intermediate, nor does it distinguish between the *syn* and *anti* isomers of the Criegee intermediates, which may have different decomposition rates and yields. Despite its potential shortcomings, the mechanism was used for simplicity to provide a rough estimate of the concentration of Criegee intermediates to compare with the experimental measurements.

Under the conditions of these experiments, the model predicts a steady-state concentration of stabilized Criegee intermediates (APINBOO) of approximately 2-6 $\times 10^8$ molecules cm$^{-3}$ (Fig. 7). Based on these results,

the observed OH interference in these experiments could be explained if approximately 5% of these intermediates dissociated and produced OH radicals inside the IU-FAGE detection cell, assuming that the transmission of these stabilized Criegee intermediates through the inlet is similar to that for OH. In contrast, the MCM predicted steady-state concentration of excited Criegee radicals (APINOOA and APINOOB) in the flow tube was calculated to be on the order of $10^3$ molecules cm$^{-3}$, much less than the observed interference. Thus it is unlikely that excited Criegee intermediates are the source of the interference, but rather the interference is the result of decomposition of SCIs in the FAGE detection cell. These results are in contrast to the results of Fuchs et al. (2016), who found that in their ozonolysis experiments that the addition of $SO_2$ as a scavenger for Criegee intermediates did not affect the observed OH signal. They concluded that SCIs were not the cause of the interference in their instrument, although they could not rule out that the interference was due to the decomposition of particular SCI isomers that did not react with $SO_2$ (Fuchs et al., 2016).

If stabilized Criegee intermediates are the source of the interference in these measurements, one might have expected to observe an interference associated with the ozonolysis of isoprene even though the expected yield of OH was below the detection limit of the instrument. Simulations using the Master Chemical Mechanism suggest that the concentration of SCIs in the isoprene ozonolysis mechanism are similar to the concentration of SCIs in the α-pinene ozonolysis mechanism. Given that OH yield from the decomposition of excited CIs in the isoprene mechanism is lower than that for the α-pinene mechanism, the absence of a detectable interference in the isoprene experiments described here may suggest that the decomposition of these intermediates inside the FAGE detection cell may also be slower. Consistent with the observation that the observed interference appears to be a constant fraction of the total OH yield in these experiments independent of the ozone concentration and the turnover rate (Figs. S5 and S6), the observed interference for each alkene is likely proportional to the OH yield from the ozonolysis mechanism, with the absence of an observed interference in the isoprene experiments consistent with the lower OH yield in the isoprene ozonolysis mechanism. Additional experiments measuring the interference from the ozonolysis of isoprene is needed to resolve this issue.

*3.3.3 Interference from Criegee intermediate decomposition on LIF-FAGE calibrations.* These results also imply that using the ozonolysis of alkenes as a potential source for calibrating LIF-FAGE instruments may lead to an overestimation of the instrument sensitivity given the potential for this source to produce an interference. As mentioned above, Hard et al. (2002) observed an interference in their LIF-FAGE instrument during calibrations using the ozonolysis of trans-2-butene under high mixing ratios of both ozone (up to 28 ppm) and trans-2-butene (greater than 12 ppb). They found that the interference disappeared in the presence of 1% water vapor. Given the potentially rapid reaction of Criegee intermediates with water vapor (Chao et al., 2015), this may suggest that the

source of their interference was also the decomposition of Criegee intermediates inside their detection cell.
However, Dusanter et al. (2008) found that the instrument sensitivities derived from the ozone-alkene calibration
technique using trans-2-butene were systematically lower than those derived from the water-vapor UV- photolysis
technique, in contrast to what might be expected if the measurements from the ozonolysis technique were impacted
by an interference from Criegee intermediates. But because these ozonolysis experiments were done under humid
conditions, it is possible that the Criegee intermediates were scavenged by water vapor prior to entering the IU-
FAGE detection cell, thus minimizing the interference.  Although the rate constant of Criegee intermediates with
water vapor depends on the structure of the intermediate (Vereecken et al., 2015), future calibrations of LIF-
FAGE instruments using the ozonolysis of alkenes should be done in the presence of water vapor or another
Criegee scavenger to insure that the calibration is not influenced by this potential interference.
**4. Atmospheric Implications**
The percent interference observed in these studies (Figs. S5, S6) is similar to the ambient measurements reported
by Mao et al. (2012), who measured an interference that was approximately 50% of the total OH signal in an
environment dominated by 2-methyl-3-buten-2-ol (MBO), monoterpenes, sesquiterpenes, and related oxygenated
compounds. Novelli et al. (2014a) also found that in their measurements of OH in several forest environments the
interference accounted for 30-80% of their total OH signal during the daytime. External addition of $SO_2$ as an SCI
scavenger during some of these measurements resulted in the complete removal of the interference, suggesting
that the source of the interference was the decomposition of ambient SCIs inside their detection cell (Novelli et
al., 2016). However, the magnitude of the interference signal relative to the calibration for OH was much greater
than the expected concentration of SCIs in these environments, which was estimated to be on the order of $5\times10^4$
molecules $cm^{-3}$ (Novelli et al., 2016). One possible reason for this discrepancy is a greater sensitivity of the LIF-
FAGE instrument to the detection of ambient SCIs relative to ambient OH, perhaps due to a greater transmission
efficiency of SCIs into the FAGE detection cell. Tests to measure the sensitivity of their LIF-FAGE instrument to
the detection of SCIs relative to OH found that the transmission of $syn$-$CH_3CHOO$ through different nozzle
designs was different from the transmission of OH radicals (Novelli et al., 2016). These results suggest that the
sensitivity of their LIF-FAGE instrument may be different for the detection of SCIs and that using the calibration
factor for OH radicals to estimate the SCI concentration from the interference may not be appropriate. However,
the sensitivity of their LIF-FAGE instrument to detection of SCIs would have to be a factor of 100 greater than
that for OH based on the estimated concentration of ambient SCIs (Novelli et al., 2016). Previous measurements
on a similar LIF-FAGE instrument demonstrated that the measured OH signal is relatively insensitive to the shape
and coating of the inlet (Stevens et al., 1994). In addition, external calibrations including inlet losses were found
to be similar to internal calibrations that did not use the inlet. These results imply that heterogeneous loss of OH
radicals on the inlet is not occurring to a significant extent (Stevens et al., 1994).
Similar to that observed by Fuchs et al. (2016), extrapolating the interference observed in the experiments
presented here as a function of the ozonolysis turnover time (Fig. 6) to concentrations and turnover rates typically
observed in the atmosphere (approximately 1.5 ppb hr$^{-1}$, Hakola et al., 2012; Fuchs et al., 2016) would suggest
that the interference under ambient conditions would be near the detection limit of the IU-FAGE instrument
(approximately $4\times10^5$ molecules cm$^{-3}$). Consistent with this result, previous measurements of OH radical
concentrations by the IU-FAGE instrument in an urban environment during CalNex (California Research at the
Nexus of Air Quality and Climate Change) using the external chemical titration technique found no evidence of
an unknown interference (Griffith et al., 2016). Measurements of OH by the IU-FAGE instrument in a forested
environment in northern Michigan during and after the CABINEX (Community Atmosphere-Biosphere
Interactions Experiment) campaign using the external chemical titration technique also did not reveal any
unknown interferences (Griffith et al., 2013). However, the measurements during CABINEX were done under
relatively cool conditions and low ozone concentrations, where daytime maximum ozone mixing ratios were
approximately 30 ppb on average. The relatively cool conditions suggest that mixing ratios of BVOCs may have
been relatively low during this campaign, as mixing ratios of isoprene were less than 2 ppb. During CalNex,
maximum daytime mixing ratios of ozone were higher (approximately 50-60 ppb on average during the week and
approximately 70-80 ppb on average on the weekends), but mixing ratios of isoprene were even lower, less than
1 ppb on average, (Griffith et al., 2016), suggesting that the mixing ratio of other BVOCs may also have been
lower.
In contrast, recent measurements of OH concentrations by the IU-FAGE instrument in an Indiana forest
using the external chemical titration technique did reveal an interference that correlated with both temperature
and ozone concentrations, similar to the results of Mao et al. (2012) (Lew et al., 2017b). Compared to the
conditions during CABINEX, these measurements were done under higher daytime maximum ozone mixing ratios
(approximately 40 ppb) and relatively warmer conditions, where isoprene mixing ratios greater than 4 ppb suggest
higher mixing ratios of other BVOCs. The magnitude of the observed interference (on the order of $10^6$ molecules
cm$^{-3}$) was similar to that observed previously by other LIF-FAGE instruments (Mao et al., 2012; Novelli et al.,
2014a; 2016), and accounted for approximately 60% of the total measured OH signal on average. The known
interference from laser-generated OH varied with laser power, ambient ozone and water concentrations, but was

approximately half of the total measured interference on some days, while on other days accounted for all of the measured interference. However, similar to that observed by Novelli et al. (2016), estimates of the ambient concentration of SCIs on the order of approximately 4-5×10$^4$ molecules cm$^{-3}$ for similar environments (Percival et al., 2013; Novelli et al., 2016) suggest that the observed interference in these measurements may not be solely due to ambient SCIs unless there are other significant sources of Criegee radicals that are not accounted for in these models.

These results suggest that although SCIs may be contributing to the observed interference, there may exist an unknown interference in these measurements that also correlates with the concentration of ozone and BVOCs, perhaps due to oxidation products of BVOCs not tested in the experiments reported here (Novelli et al., 2016; Fuchs et al., 2016). Recently, Fuchs et al. (2016) reported an interference in their LIF-FAGE instrument associated with NO$_3$ radicals, although the exact mechanism of the interference remains unknown. These results suggest that there may be other potential interferences associated with the technique in addition to the decomposition of SCIs that involve complex homogeneous and/or heterogeneous mechanisms inside the FAGE detection cell.

## 5. Summary and Conclusions

Measurements of OH concentrations produced from the ozonolysis of α-pinene, β-pinene, and ocimene revealed a potential interference associated with the Indiana University LIF-FAGE instrument. The observed interference did not appear to be laser generated and was independent of the ozonolysis reaction time. Addition of acetic acid resulted in measured OH concentrations that were similar to measurements when the interference was subtracted, suggesting that the source of the interference in these experiments involved the decomposition of stabilized Criegee intermediates inside the IU-FAGE detection cell. Further measurements and modeling will be needed for a wider variety of alkenes in order to confirm these results.

The interference appeared to increase with the length of the inlet in the low pressure region of the detection cell, suggesting that the interference depends on the reaction time in the detection cell. However, increasing the pressure in the detection cell by decreasing the flow rate did not significantly increase the observed interference except for the longest inlet. This may suggest that the increase in the observed interference with the length of the inlet may be the result of increased collisions of the stabilized Criegee intermediates inside the detection cell leading to the formation of OH rather than the result of an increase in reaction time. Additional experiments will be needed to confirm these results. To minimize this interference and other unknown

interferences, future measurements of OH by the IU-FAGE instrument will test different detection cell designs to
minimize any observed ambient interferences though changes in both reaction time and potential surface
collisions, while also maintaining the quality of the ambient OH measurement.
Regardless, future ambient measurements by the IU-FAGE instrument will incorporate the external OH
titration technique to quantify these and other potential unknown interferences. Because of differences in design
(geometry, cell pressure, flow, etc.) these interference measurements may not apply to other LIF-FAGE
instruments. However, it is recommended that future OH measurements using the LIF-FAGE technique
incorporate an external OH titration scheme or some other method to quantify potential artefacts.
Acknowledgements: This work was supported by the National Science Foundation, grants AGS-1104880 and
AGS-1440834.  We would like to thank Sebastien Dusanter for helpful comments on the manuscript.

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

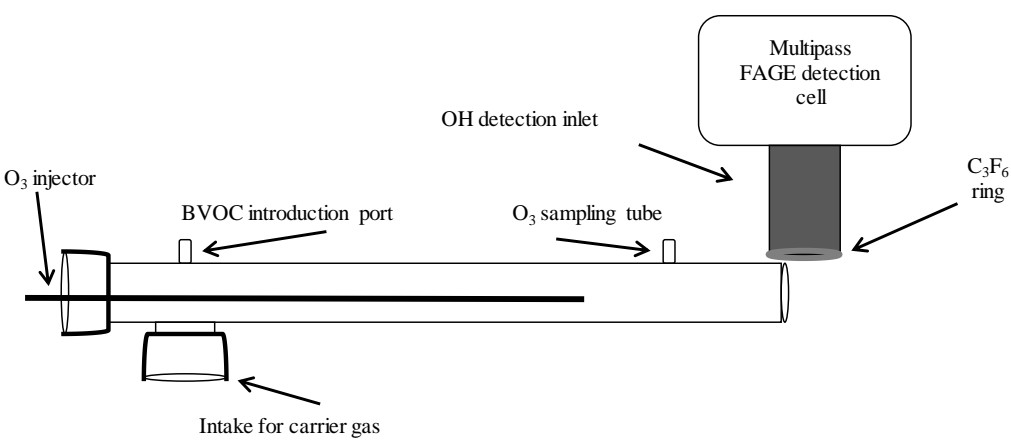

**Figure 1:** Schematic of the atmospheric pressure flow system used in this study.

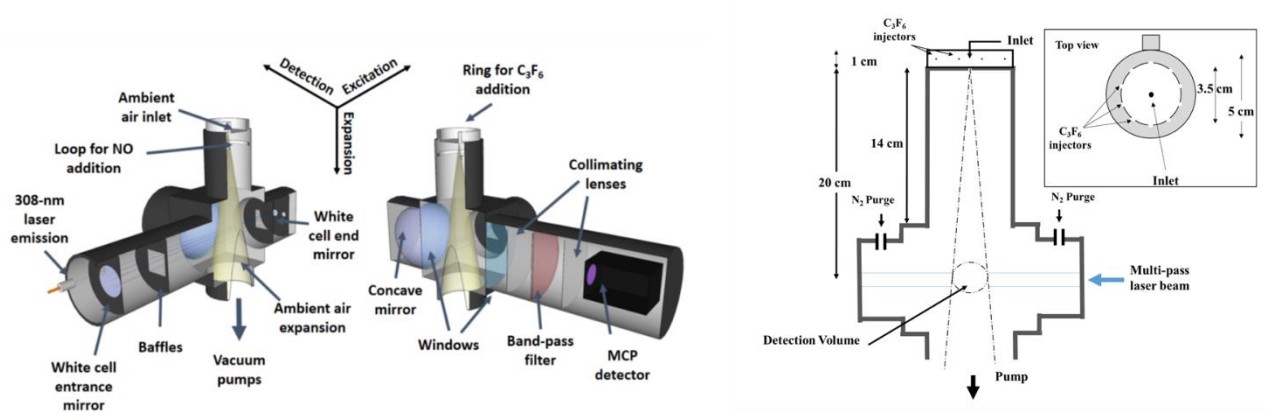

**Figure 2.** Schematic of the IU-FAGE sampling/excitation axis (left) and cross section with dimensions for the medium inlet configuration (right).

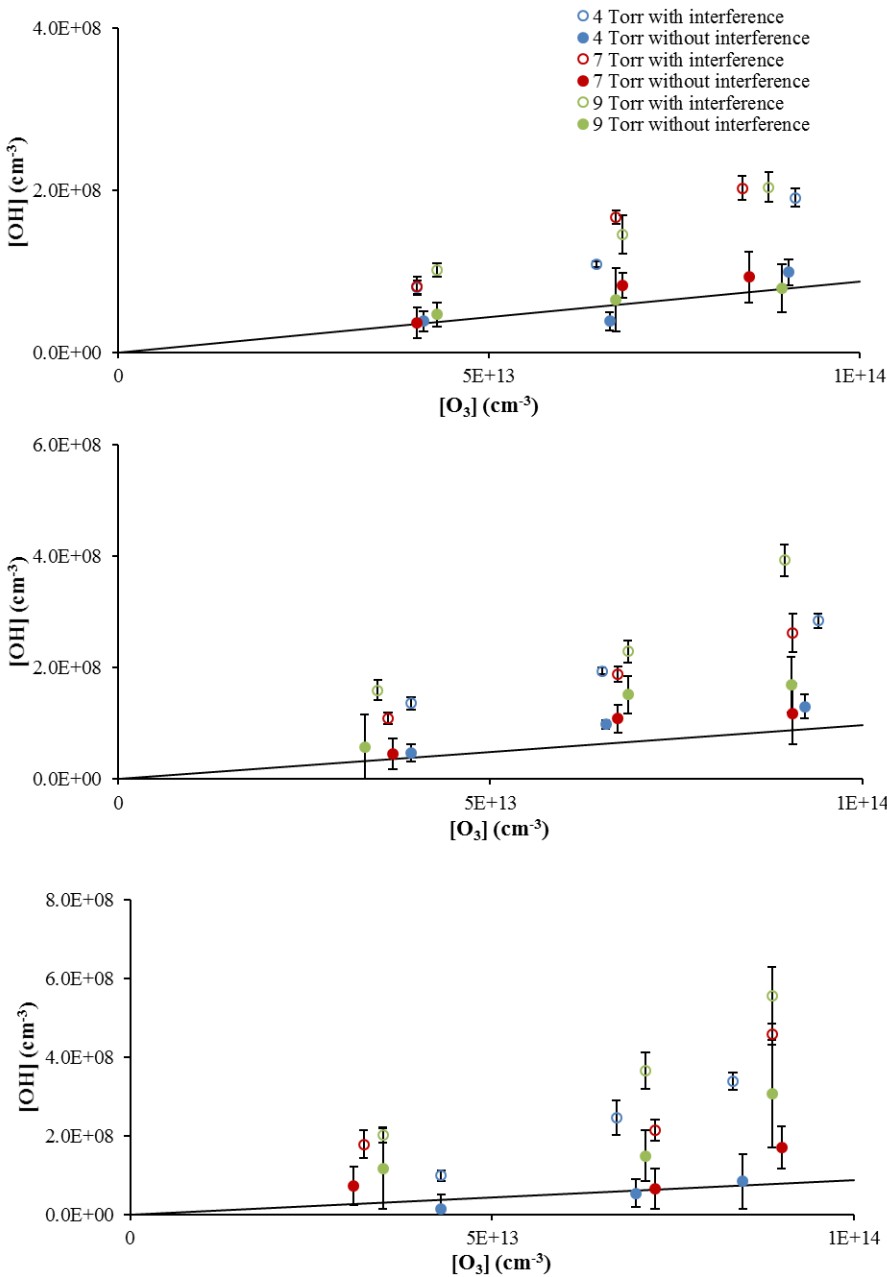

**Figure 3.** OH concentrations from α-pinene ozonolysis at three cell pressures with and without the interference using the 0.6 mm diameter nozzle and the short (top), medium (middle), and long (bottom) inlet lengths with α-pinene concentrations of approximately $3 \times 10^{12}$ cm$^{-3}$. Error bars indicate the precision of the measurement (1σ). Lines indicate the expected steady-state OH concentration based on published values of the OH yield (see text).

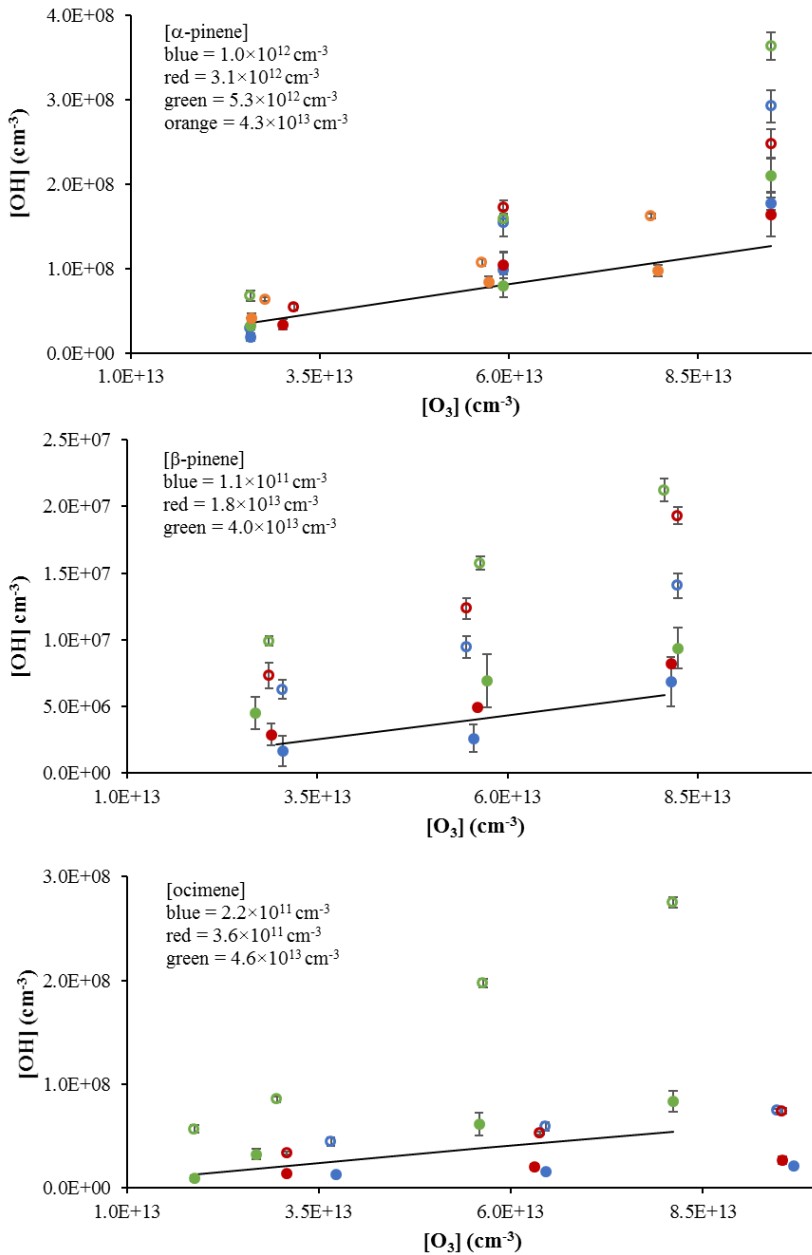

**Figure 4**. OH concentrations using the short inlet from the ozonolysis of α-pinene (top), β-pinene (middle), and ocimene (bottom). Open circles indicate measurements with the interference, filled circles indicate measurements without the interference. Colors indicate estimated concentrations. Solid lines reflect the expected OH radical yield (see text). Error bars reflect the measurement precision (1σ).

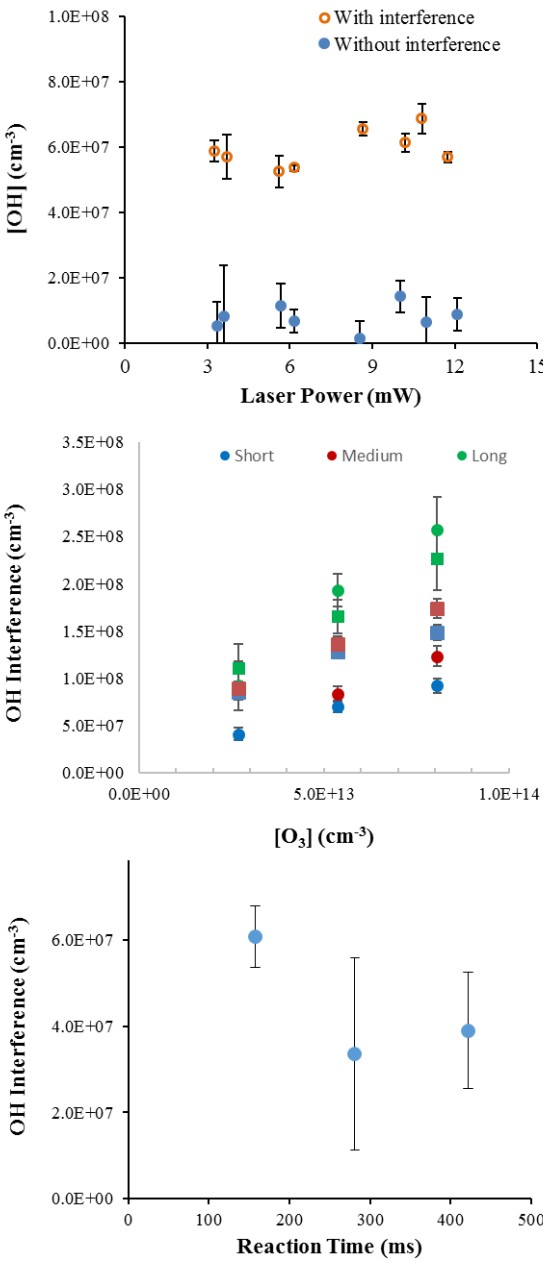

**Figure 5.** OH concentrations from ocimene ozonolysis with varied laser power at an ozone mixing ratio of 2 ppm and ocimene concentration of approximately $3 \times 10^{13}$ cm$^{-3}$ (top). OH interference during $\alpha$-pinene (approximately $3 \times 10^{12}$ cm$^{-3}$, circles) and ocimene (approximately $3 \times 10^{13}$ cm$^{-3}$, squares) ozonolysis based on ozone concentration and inlet length (middle). OH interference measurements during ocimene ozonolysis as a function of reaction time with an ocimene concentration of approximately $3 \times 10^{13}$ cm$^{-3}$ (bottom).

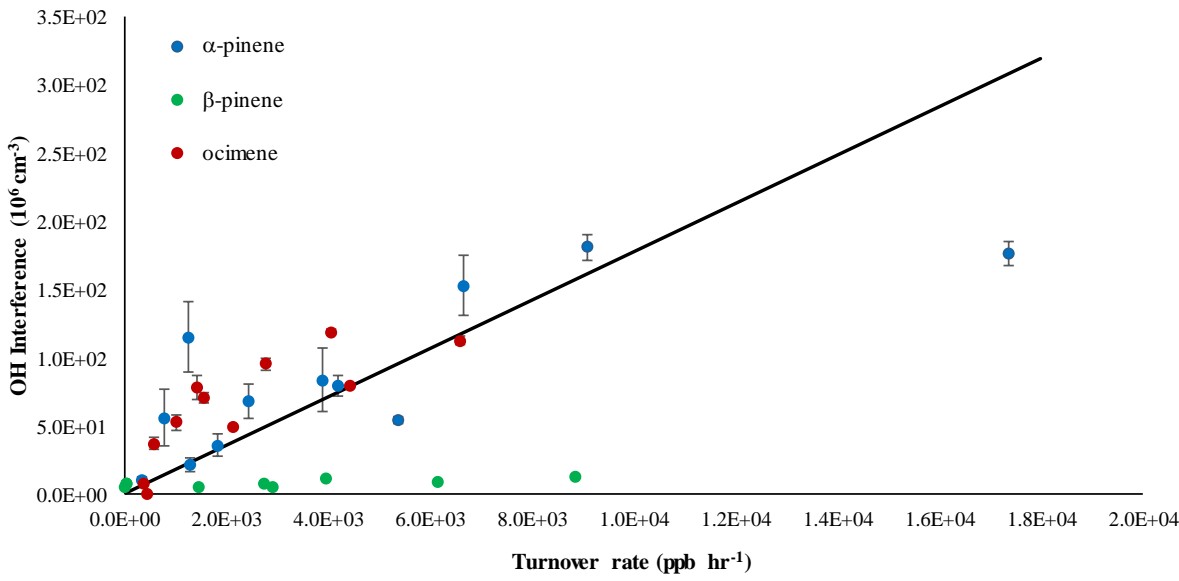

**Figure 6.** Interference signal as a function of turnover rate for the ozonolysis of α-pinene, β-pinene, and ocimene using the short inlet. Solid line reflects the slope observed by Fuchs et al. (2016).

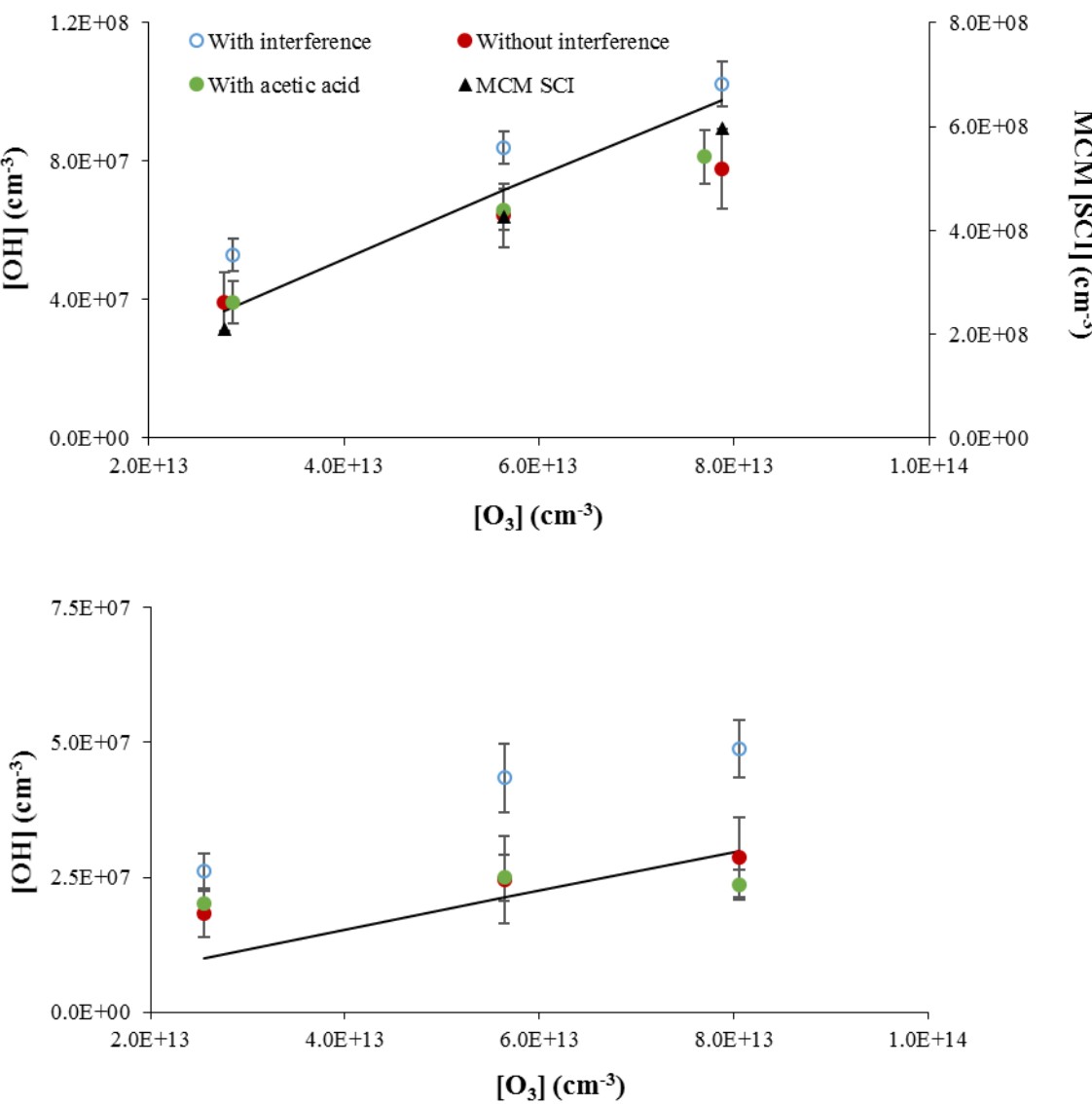

**Figure 7.** Measurements of OH concentrations from the ozonolysis of $\alpha$-pinene (approximately $1 \times 10^{12}$ cm$^{-3}$, top) and ocimene (approximately $4 \times 10^{11}$ cm$^{-3}$, bottom). Open symbols are measurements including the interference, filled red circles are the resulting OH measurements when the interference determined by $C_3F_6$ addition is subtracted, and the filled green circles are the OH measurements when acetic acid is added to the flow tube. The lines indicate the expected OH concentration from published OH yields. The black triangles in the top plot reflect the predicted concentration of stabilized Criegee intermediates by the Master Chemical Mechanism (MCM SCI) (see text).