# Peer review of "Measurements of a potential interference with laser-induced fluorescence measurements of ambient OH from the ozonolysis of biogenic alkenes"

_Atmospheric Measurement Techniques, 2017_

## Referee Comment (RC1) · Anonymous Referee #1 · 25 Jul 2017

I was very interested to review this manuscript which falls well within the scope of AMT and reports on the extremely important subject of potential OH artefacts in LIF instruments used for ambient OH measurements. Although other LIF groups have reported their findings from their own LIF instruments (Mao et al, Novelli et al and Fuchs et al), it is very important for the experiments presented in this manuscript to be conducted and published by every LIF group involved in ambient OH detection. In this manuscript there are a few key experimental details missing in places, particularly

the Inlet Pre Injector parameters, which need to be included in the revised manuscript (discussed below). I also have some concerns over the experimental approach (which will likely be resolved once further experimental detail is provided) and the presentation of the results could be improved upon. Notwithstanding, once these changes are made I fully recommend publication in AMT.

Specific comments

Abstract: In general there needs to be further specific details on the key findings included in the abstract.

Line 16: 'several BVOCs.' these should be named in here

Line 18: 'an interference under high ozone and BVOC concentrations was observed..' It is important to state the level of the interference in the abstract. I understand that this level varied with [O3] and [BVOC] and BVOC type, but I suggest reporting the maximum interference observed and giving the concentration of the pre-cursors for a particular experiment. It is also important to state here the anticipated interference under ambient conditions somewhere in the abstract.

Pg 2, lines 28-30: Mention specific chemical condition under which the measurements were made.

Pg 4, lines 10-11: What is the motivation for choosing these specific BVOCs? Has ocimene been observed at appreciable levels in forested environments?

Pg 5, line 2: Although no OH or artefact signal was observed during experiments conducted with isoprene and MBO, the experimental conditions, i.e. the concentration of isoprene and MBO (and ozone concentration if different from the other ozonolysis experiments) should be added to the experimental section.

Section 2.1: It is unclear which laser was used for the experiments detailed in this manuscript.

How do the pressures tested in these experiments compare to those typically employed during ambient measurements?

Section 2.2: The authors need to provide additional details on their chemical titration scheme. Specifically, what is the total flow rate through the chemical titration section of the instrument? What is the concentration of C3F6 added (in molecule cm-3)? What is the residence time of gas in the chemical titration section? These parameters are important as I worry that if only âľ¿90% of a point OH source (from the calibration wand) is removed by the scavenger, then even less OH generated via a steady state source (O3+BVOC) will be removed and this could lead to a bigger percentage of the OH signal observed being assigned as an interference than is necessarily the case. Other experimental results presented, such as the agreement of the OH yield with literature values, and the variation in the magnitude of the interference with inlet length do suggest that the amount of scavenger injected is sufficient to remove a steady state source of OH, but it is important to demonstrate this absolutely. The authors could consider presenting results from a simple kinetic model which includes the main OH source (O3+alkene) and sink reactions (OH+alkene, OH+C3F6 reaction..), run over the residence time in the chemical titration section, to demonstrate this?

Pg 8, lines 27, 28: There does seem to be some trend with beta-pinene concentration?

Pg 9, line 15: As well as reflecting the 'higher reactivity' of the mono-terpenes with ozone compared to isoprene and MBO, important also (to the real OH signal) is relative reactivity of BVOC+O3 vs BVOC+OH (and the OH yield from ozonolysis). All should be mentioned as possible reasons for the lack of real OH signal observed. I am a little surprised that no OH signal was observed during these experiments even with the shortest inlet given the limit of detection stated in section 2.1.

Is the concentration of SCI in the isoprene+O3 and MBO+O3 experiments estimated to be lower than during the monoterpene+O3 experiments? The rate coefficients, kisop+o3 and kbetapinene+o3 are similar.

[Figure]

Pg 10, lines 27- 30: It is interesting/perplexing that the artefact signal is actually lower when the medium inlet is used than when the shorter inlet is used. In light of previous results (Fuchs et al. 2016) which demonstrated a dependence of the magnitude of the artefact signal on cell residence time a comment on the lack of trend in level of interference and inlet length is needed here.

Section 3.3: The working hypothesis on the identity of the observed OH interference is that it derives from the decomposition of a SCI. Were any experiments conducted with alkene concentration in excess? Under these conditions the concentration of the SCI would be maximised, whilst the concentration of externally generated OH from ozonolysis would be small, meaning that the artefact signal should be readily distinguishable from a real OH signal?

Pg 12, lines 1 – 17: What was the concentration of acetic acid added to the flow-tube? Would any loss of OHss by reaction with acetic acid be expected given the residence time?

Pg 12, lines 22 – 24: 'Based on these results, the observed OH interference in these experiments could be explained if approximately 5% of these intermediates dissociated..' Does this then effectively disprove the hypothesis that the transmission efficiency of SCI vs OH through the pinhole is substantially different? A comment on transmission efficiency assumed for these lab results vs transmission efficiency estimated from field results (and the implications of these differences) would be welcomed in the revised manuscript.

Technical corrections

Pg 2, line 26: 'their' to 'Penn State'

Pg 4, line 15: add ' with sliding injector' after 'flow tube' so the later discussion on the injector is easier to follow.

Pg 4, line 17: Define 'IU-FAGE'

Pg 6, line 20: add ' compared to chemical modulation' after 'spectral modulation'?

Pg 6, line 21: change 'reflect' to 'can reveal'?

Pg 6, line 25: is this 3 – 5 sccm of 1% C3F6 in N2 or 3 – 5 sccm pure C3F6?

Pg 9, line 7: ±0.9 is a very large error. Is this correct?

Pg 10, line 19: Define 'turnover time'

Figures: Stick to [O3] in molecule cm-3 or ppm.

Figure 3 – 7: Axes should be rescaled and legends should be made more self-explanatory. It took me a while to understand what 'Pcell 4' actually represented.

Figure 3 & 4: It isn't clear to me why the OH yield from the ozonolysis reactions and the OH signal without scavenger are on the same graph? They are two distinct results that just happened to have been determined in the same experiment. I am struggling to suggest a better way to present the results, but maybe the authors could critically review these figures before final publication?

Figure 4: Include a legend on this figure that clearly states the VOC concentration for the different experiments, e.g. green = x cm-3

Figure 5: it is not obvious to me why these three panels are grouped together? The recommendations for improving the figures above should be considered for the figures included in SI also.

---

## Referee Comment (RC2) · Anonymous Referee #2 · 10 Aug 2017

This manuscript describes tests performed on the LIF-FAGE instrument in use in the Indiana University to assess the presence of some interference species in the OH radical measurement. The tests performed within this study follow previous work from different LIF-FAGE groups and focus on the ozonolysis of different alkenes. The study shows that an interfering signal is observed during the ozonolysis of specific BVOC but extrapolating these results to ambient concentrations suggest that the interference coming from ozonolysis of unsaturated VOCs will have a negligible impact. I think the

topic is of interest as it has been shown how different LIF-FAGE instruments (all with different instrumental parts, flows, etc) react differently to interfering species underlining the need of a characterization for each and every instrument. The manuscript is well written and structured though, in my opinion, it addresses the issue of the interference species insufficiently in-depth and it needs a more extensive characterization and analysis of the results.

A first general topic that needs to be address is a characterization of the titration unit used for the investigation of the interfering species. The literature cited when referring to the titration unit does not appear to give a full characterization of the device. As this paper focuses on the interference on the OH radical measurement and as the titration is currently in use in field campaigns, this would be the appropriate study to include the details about the titration unit such as losses on walls, plots with titration efficiency at ambient pressure and low pressure within the detection axis, dependency on the different parameters such as air flow, mixing volume, etc. This could be added in section 2.2.

The title of the manuscript is misleading. Neither in the abstract or in the conclusion OH radical yields from ozonolysis of selected alkenes are discussed as such. The study focuses mainly on the impact of the interference on the instrument rather than providing new insight in the OH yield. Therefore I feel there should be a more extensive analysis or discussion of possible interfering species. The interference from NO3 as described by Fuchs et al. (2016) is mentioned but this would be the chance to actually perform tests in the laboratory to see how much this particular LIF-FAGE instrument is affected by it. The same is valid for other species as the study from Ren et al. (2004) applies to the LIF-FAGE instrument to which the tests were done. This is a good study to advance the knowledge on the interference species within the OH measurement of the LIF-FAGE by trying new/different possible trace gases.

Section 3.3 needs a lot more explanation and clarification: - It is not clear what the hypothesis of the authors is. The first paragraph of this section distinguishes between
excited and stabilized Criegee intermediates mentioning that the first produce OH at short times (how short?) and the second at longer time (how long?), Are the authors arguing that the OH radicals they observed in the flow tube are only coming from the excited Criegee intermediates? If that is the case it should be stated explicitly. Though, it would be hard to explain how the OH would be formed within the instrument if the stabilized Criegee intermediates would not decompose within the flow tube. This assumption needs to be check carefully as several studies conclude that the unimolecular decomposition rate of stabilized Criegee intermediates is rather fast (Smith et al., 2016;Chhantyal-Pun et al., 2016).

- The comparison of the signals as shown in Figure 7 is only valid if the OH formed within the flow tube originates from excited Criegee intermediates only. Formulas (for example, TOT signal = OHFlowTube (OHExcitedCriegee + OHStabilizedCriege + OHInterference) explaining clearly the expected component of every signal should be added to avoid confusion. If the OH observed in the flow tube also originates from stabilized Criegee intermediates it would not be possible to compare full red and full green symbols as the injection of acetic acid would remove a source, within the flow tube, for the OH observed.

- The comparison with the MCM 3.2 needs a lot more detail. The MCM mechanism as is does not include the chemistry needed to do a proper comparison, e.g. which unimolecular rate coefficient was used for the decomposition of the stabilized Criegee intermediate? On which assumptions/studies is the rate coefficient based? How do the authors deal with the fact that one of the two excited Criegee intermediates in the MCM does not decompose forming a stabilized Criegee intermediates (APINAOO)? How is the speciation of the 4 SCI formed from a-pinene treated (e.g. syn vs anti chemistry, relative yields, different unimolecular channels)? Are there additional losses included in the model for the SCI?

Figures need to be revised. In particular legends are not easy to understand rendering the message of the figure not very clear. I would recommend publication in AMT once
these general points are addressed.

Specific comments:

Title: As suggested above, the title needs to be change as the focus of the study is the study of the LIF-FAGE interference. The yield of OH radicals from the ozonolysis of BVOCs does not seem to be the main topic of the manuscript.

Page 2, Lines 9 to 16: The OP3 field campaign results (Whalley et al., 2011) needs to be added.

Page 2, Lines 21 to 27: Tests were done on a specific instrument, it needs mentioning.

Page 3, Line 20: The OH radical concentration measured with the LIF-FAGE agrees with the measurements performed with two CIMS instruments.

Page 3, Line 28 to 30: Here the text is misleading. Criegee intermediates decompose forming OH at low pressure and ambient pressure. Several experimental studies are now available proving the decomposition path and suggesting a rate (Smith et al., 2017;Kidwell et al., 2016;Fang et al., 2016) plus extensive theoretical material (see tables in Vereecken and Francisco (2012)).

Page 4, Line 21: Is this reaction time measured or calculated?

Page 4, Line 24: The majority of the tests described in this study include data points collected at 3 ozone values (I assume 1, 2 and 3 ppm). Was it not possible to explore a larger range of ozone values that would make the fit more robust? The accuracy of the ozone measurement needs to be added.

Page 4, Lines 26 on: Was any measurement done with, for example, a GC instrument to compare the calculated concentration of VOCs with the measured one? How were the losses on walls accounted for? What is the error on the estimated concentration? What are the concentrations for isoprene and MBO? A table summarizing the different experiments, at which conditions they were performed and the amount of interference
observed would be helpful.

Page 5, Line 12 and 16: It is not clear which laser is the new one as both lasers have the same identification number.

Page 5, Line 20: Is there a particular reason to use a 12 m fiber in the laboratory? Do the authors expect a dependency of the interference on the length of the optical fiber?

Page 6, Line 1 to 2: How long does it take for the OH concentration to stabilize?

Page 6, Line 5: Why are the experiments performed in N2 and not synthetic air?

Page 6, Line 13: "the limit of detection was approximately between...". Summarizing the different sensitivity of the instrument for different parameters and inlet configuration in a table rather than a plot would be helpful and the error on the values should be stated.

Page 6, Section 2.2: As underlined above, a more in detailed characterization of the titration unit with figures of the scavenging experiments, wall losses values, dependency of the OH scavenging on the flow of air sampled, and on the mixing volume, etc., need to be added.

Page 7, Line 10: For which conditions was the steady state reached in 20 ms? The figures need to be self-explanatory. More text needs to be added in the figure caption together with a clearer legend.

Page 7, Line 16: For consistency: kO3+VOC and kOH+VOC.

Page 7, Line 18: How much are kwall and kOH+O3x[O3] for the experiments performed in this study?

Page 7, Line 18: "measured as describe in previous work by Handen et al., 2014..".

Page7, Line 23: In figures 3 and 4, why are the data point with and without C3F6 added for a certain pressure in the cell at different ozone values? Those points are
taken consequently or? Is the variation in the ozone due to instability of the ozone generator? Is one data point in the plot the average of a single experiment or the average of the repetition of different experiments performed at the same conditions? What kind of fit is applied? Is it weighted on the errors? Does it account for errors on both x and y axis?

Page 8, Line 9: The fit showed in the central and bottom panel of figure S3 hardly represents the data. Was here used a different fit? Is there an explanation for the extremely higher values for the interference for the long inlet with the 1 mm pinhole (bottom panel figure S3) compared to the values observed for the same inlet with the 0.6 mm pinhole (bottom panel figure 3)? The values for the other two inlet length did not show a drastic variation between the 2 different pinholes. Could this be related to a smaller drop in sensitivity observed between medium and long inlets with the 1 m pinhole compared to the 0.6 mm one?

Page 8, Line 14: "The different pressures in these experiments..."

Page 8, Line 15: Remove likely

Page 8, Lines 18 to 20: The longer inlet will also increase the OH losses so it is still not clear why with the longer inlet there is an increase of interference when increasing the pressure.

Page 8, Lines 20 to 21: A legend should be added to figure 4 together with the errors on the concentrations of the BVOC.

Page 8, Lines 27 to 28: By looking at figure 4 central panel it is possible to observe a trend with higher concentration of interference for higher concentration of  $\beta$ -pinene.

Page 8, Line 30: Which Ocimene isomer was used during the experiments?

Page 9, Line 9: Remove likely.

Page 9, Line 14: Was the expected steady state concentration of OH radicals for the
condition of the experiments calculated? Rate coefficient of Isoprene with O3 is less than a factor of 2 slower than the one with  $\beta$ -pinene and the tabulated OH yields are similar for both (~0.25) so it is not clear why there would not be any detectable OH signal especially at the highest ozone concentration.

Page 10, Line 10: To which experiments does "These experiments" refer to?

Page 10, Line 15: Use molecules cm-3 for the x axis as in the previous plots. Here it would be interesting to also have in a figure the amount of interference from  $\beta$ -pinene and ocimene.

Page 10, Line 18: Ocimene and  $\beta$ -pinene should be added to Figure 6 to see if they lie on the same line as it is shown in Fuchs et al. (2016). The possible explanation about the large scatter observed in the data should also be given. How does the plot look like for longer inlets?

Page 10, Line 24 on: Here a table including also  $\beta$ -pinene (or is there a reason not to list it?) could substitute the plots.

Page 11, Line 3 on: The last paragraph is difficult to understand. I see what point the authors want to make (although I am not sure this is the appropriate place in the manuscript to make this point) but the text could benefit from rephrasing.

Page 11, Lines 11 to 15. Criegee intermediates can also be formed directly in a stabilized form from non-endocyclic double bonds. It would also be helpful at this stage to give an estimate of the time scale where these CI give OH, i.e. stabilized CI of the order of milliseconds, collisional stabilisation is of the order of 108 s-1, prompt decomposition is thus at even faster rates, also implying a very low steady state concentration of excited CI. The OH concentrations are not in steady state because the excited CI are in steady state: these two species have different formation and destruction timescales, where excited CI will reach steady state concentration orders of magnitude faster than OH. It also needs to be specified that the steady state is reached only within the flow AMTD
tube and not inside the instrument. In the conclusions the authors also suggest that SCI decomposition may not be constant throughout the detection cell, e.g. more SCI migrating to the walls and only then undergoing decomposition. Such effects should also be discussed in more detail at the start so a complete kinetic model is available prior to interpreting the results.

Page 12, Line 5: Was it not possible to try different SCI scavengers like water and/or SO2?

Page 12, line 24 to 26. The modeling of the concentration of excited CI should go to the section at the end of page 11 to further strengthen the assertion that excited CI cannot be the source of the interference. This should also be stated explicitly.

Page 12, Lines 28 "Criegee..".

Page 13, Lines 2 to 3: Chao et. al 2015 measured a fast rate coefficient for CH2OO with water dimers but it is not a good idea to generalize this rate for all the Criegee intermediates as several studies (see Vereecken et al. (2015) and citations therein) have shown that the rate with water and water dimers will strongly depend on the structure of the Criegee intermediates.

Page 13, Line 6: The authors discuss the potential impact of using alkene ozonolysis on FAGE calibration. It could be beneficial to separate that out in a separate paragraph or even a separate section.

Page 14, Line 14: The Isoprene concentration is mentioned as indicative on the likelihood of interferences in this specific LIF-FAGE instrument but as no interference was observed for concentrations way larger than what observed in the field it does not seem to be the appropriate parameter.

Page 14, Line 23: What was the percentage of the interference compared to the "real" atmospheric OH? How much was the known ozone interference?

Page 14, Lines 29 to 31: Any hypothesis on what could be the cause for the interfer-
ence?

Page 15, Lines 8 to 10: This sentence is correct if contemporary to the addition of Acetic acid also C3F6 was injected and no OH signal was observed. Was this the case?

Page 15, Lines 18 to 20: Trying to minimize any interference is important, but isn't this specific interference actually, probably, not so relevant in field campaigns? Is there going to be a gain in modifying the instrument for a negligible interference risking losing in sensitivity or encountering different problems? Are there tests showing that the modification improve the quality of the OH measurement?

References:

Chhantyal-Pun, R., Welz, O., Savee, J. D., Eskola, A. J., Lee, E. P. F., Blacker, L., Hill, H. R., Ashcroft, M., Khan, M. A. H. H., Lloyd-Jones, G. C., Evans, L. A., Rotavera, B., Huang, H., Osborn, D. L., Mok, D. K. W., Dyke, J. M., Shallcross, D. E., Percival, C. J., Orr-Ewing, A. J., and Taatjes, C. A.: Direct Measurements of Unimolecular and Bimolecular Reaction Kinetics of the Criegee Intermediate (CH3)2COO, The Journal of Physical Chemistry A, 10.1021/acs.jpca.6b07810, 2016.

Fang, Y., Liu, F., Klippenstein, S. J., and Lester, M. I.: Direct observation of unimolecular decay of CH3CH2CHOO Criegee intermediates to OH radical products, The Journal of Chemical Physics, 145, 044312, doi:http://dx.doi.org/10.1063/1.4958992, 2016.

Fuchs, H., Tan, Z., Hofzumahaus, A., Broch, S., Dorn, H. P., Holland, F., Künstler, C., Gomm, S., Rohrer, F., Schrade, S., Tillmann, R., and Wahner, A.: Investigation of potential interferences in the detection of atmospheric ROx radicals by laser-induced fluorescence under dark conditions, Atmos. Meas. Tech., 9, 1431-1447, 10.5194/amt-9-1431-2016, 2016.

Kidwell, N. M., Li, H., Wang, X., Bowman, J. M., and Lester, M. I.: Unimolecular dissociation dynamics of vibrationally activated CH3CHOO Criegee intermediates to OH

AMTD
radical products, Nat Chem, advance online publication, 10.1038/nchem.2488

Ren, X., Harder, H., Martinez, M., Faloona, I. C., Tan, D., Lesher, R. L., Di Carlo, P., Simpas, J. B., and Brune, W. H.: Interference Testing for Atmospheric HOx Measurements by Laser-induced Fluorescence, J. Atmos. Chem., 47, 169-190, 10.1023/b:joch.0000021037.46866.81, 2004.

Smith, M. C., Chao, W., Takahashi, K., Boering, K. A., and Lin, J. J.-M.: Unimolecular Decomposition Rate of the Criegee Intermediate (CH3)2COO Measured Directly with UV Absorption Spectroscopy, The Journal of Physical Chemistry A, 10.1021/acs.jpca.5b12124, 2016.

Smith, M. C., Chao, W., Kumar, M., Francisco, J. S., Takahashi, K., and Lin, J. J.-M.: Temperature-Dependent Rate Coefficients for the Reaction of CH2OO with Hydrogen Sulfide, The Journal of Physical Chemistry A, 10.1021/acs.jpca.6b12303, 2017.

Vereecken, L., and Francisco, J. S.: Theoretical studies of atmospheric reaction mechanisms in the troposphere, Chemical Society Reviews, 41, 6259-6293, 10.1039/c2cs35070j, 2012.

Vereecken, L., Glowacki, D. R., and Pilling, M. J.: Theoretical Chemical Kinetics in Tropospheric Chemistry: Methodologies and Applications, Chemical Reviews, 10.1021/cr500488p, 2015.

Whalley, L. K., Edwards, P. M., Furneaux, K. L., Goddard, A., Ingham, T., Evans, M. J., Stone, D., Hopkins, J. R., Jones, C. E., Karunaharan, A., Lee, J. D., Lewis, A. C., Monks, P. S., Moller, S. J., and Heard, D. E.: Quantifying the magnitude of a missing hydroxyl radical source in a tropical rainforest, Atmos. Chem. Phys., 11, 7223-7233, 10.5194/acp-11-7223-2011, 2011.

---

## Author Comment (AC1) · 9 Oct 2017

**Response to Anonymous Referee #1**

We would like to thank the reviewers for their efforts in reviewing this manuscript, and we feel that the manuscript is much stronger with the suggested changes. Below are detailed responses to their comments, which are highlighted in italics.

*I was very interested to review this manuscript which falls well within the scope of AMT and reports on the extremely important subject of potential OH artefacts in LIF instruments used for ambient OH measurements. Although other LIF groups have reported their findings from their own LIF instruments (Mao et al, Novelli et al and Fuchs et al), it is very important for the experiments presented in this manuscript to be conducted and published by every LIF group involved in ambient OH detection. In this manuscript there are a few key experimental details missing in places, particularly the Inlet Pre Injector parameters, which need to be included in the revised manuscript (discussed below). I also have some concerns over the experimental approach (which will likely be resolved once further experimental detail is provided) and the presentation of the results could be improved upon. Notwithstanding, once these changes are made I fully recommend publication in AMT.*

*Specific comments*

*Abstract: In general there needs to be further specific details on the key findings included in the abstract.*

*Line 16: 'several BVOCs..' these should be named in here*

We have added the names of each of the BVOCs measured to the abstract as suggested.

*Line 18: 'an interference under high ozone and BVOC concentrations was observed..' It is important to state the level of the interference in the abstract. I understand that this level varied with [O3] and [BVOC] and BVOC type, but I suggest reporting the maximum interference observed and giving the concentration of the pre-cursors for a particular experiment. It is also important to state here the anticipated interference under ambient conditions somewhere in the abstract.*

As suggested, we have added the average level of the interference observed and the range of precursor concentrations to the abstract. We have also added a statement concerning the anticipated interference under ambient conditions.

*Pg 2, lines 28-30: Mention specific chemical condition under which the measurements were made.*

We have added that the measurements were made under varying concentrations of of $H_2O$, $O_3$, CO, HCHO, NO, and $NO_2$ as described in Schlosser et al., 2007.

*Pg 4, lines 10-11: What is the motivation for choosing these specific BVOCs? Has ocimene been observed at appreciable levels in forested environments?*

The BVOCs chosen represent major monoterpene emissions ($\alpha$- and $\beta$-pinene) as well as a frequent emission (cis-ocimene) (Guenther et al., Atmos. Environ., 28, 1197-1210, 1994), in addition to isoprene and MBO. This has been clarified.

*Pg 5, line 2: Although no OH or artefact signal was observed during experiments conducted with isoprene and MBO, the experimental conditions, i.e. the concentration of isoprene and MBO (and ozone concentration if different from the other ozonolysis experiments) should be added to the experimental section.*

We have added the approximate concentrations of both isoprene and MBO used in these experiments to the experimental section as suggested.

*Section 2.1: It is unclear which laser was used for the experiments detailed in this manuscript.*

This has been clarified in the revised manuscript.

*How do the pressures tested in these experiments compare to those typically employed during ambient measurements?*

We have clarified the normal operating pressures and inlet lengths typically used during previous ambient measurements as suggested.

*Section 2.2: The authors need to provide additional details on their chemical titration scheme. Specifically, what is the total flow rate through the chemical titration section of the instrument? What is the concentration of C3F6 added (in molecule cm-3)? What is the residence time of gas in the chemical titration section? These parameters are important as I worry that if only 90% of a point OH source (from the calibration wand) is removed by the scavenger, then even less OH generated via a steady state source (O3+BVOC) will be removed and this could lead to a bigger percentage of the OH signal observed being assigned as an interference than is necessarily the case. Other experimental results presented, such as the agreement of the OH yield with literature values, and the variation in the magnitude of the interference with inlet length do suggest that the amount of scavenger injected is sufficient to remove a steady state source of OH, but it is important to demonstrate this absolutely. The authors could consider presenting results from a simple kinetic model which includes the main OH source (O3+alkene) and sink reactions (OH+alkene, OH+C3F6 reaction), run over the residence time in the chemical titration section, to demonstrate this?*

We have expanded the description of the chemical titration scheme in this section, including a schematic diagram of the injector ring in Figure 2. We have also included estimates of the concentration of $C_3F_6$ added and the residence time in the titration region as suggested. We have also provided results of a simple kinetic model, which shows that the amount of $C_3F_6$ added and the residence time in the titration region, is enough to reduce the steady-state concentration of OH from the ozonolysis reactions to below the detection limit of the instrument for the majority of the experiments described here. However, it is possible that for some of the high concentration experiments, the amount of $C_3F_6$ added may not have been sufficient to reduce the steady-state concentration of OH to below the detection limit, especially for the ocimene experiments due to the high reactivity of ocimene with ozone. However, the model simulations suggest that even for these high concentration experiments the remaining steady-state OH concentrations represented less than 10% of the observed interference. This has been clarified in the revised manuscript.

*Pg 8, lines 27, 28: There does seem to be some trend with beta-pinene concentration?*

We have added a statement indicating that there appears to be a trend in the measured OH yield with increasing β-pinene concentration.

*Pg 9, line 15: As well as reflecting the 'higher reactivity' of the mono-terpenes with ozone compared to isoprene and MBO, important also (to the real OH signal) is relative reactivity of BVOC+O3 vs BVOC+OH (and the OH yield from ozonolysis). All should be mentioned as possible reasons for the lack of real OH signal observed. I am a little surprised that no OH signal was observed during these experiments even with the shortest inlet given the limit of detection stated in section 2.1.*

We have clarified that the lower expected steady-state OH concentration in the ozonolysis of isoprene and MBO are lower due to the relative reactivity with ozone and OH as well as the overall OH yield as suggested. We have also performed simulations that show that the expected OH concentration in the isoprene experiments were approximately 50 times lower than that for the α-pinene experiments, consistent with a lower steady-state OH concentration estimated using Equation 1, and near or below the detection limit of the instrument.

*Is the concentration of SCI in the isoprene+O3 and MBO+O3 experiments estimated to be lower than during the monoterpene+O3 experiments? The rate coefficients, kisop+o3 and kbetapinene+o3 are similar.*

As pointed out by the reviewer, simulations using the Master Chemical Mechanism suggest that the concentration of SCIs in the isoprene + $O_3$ experiments is similar to that in the α-pinene + O3 experiments. This may suggest that a similar interference should have been observed during the isoprene experiments as was observed during the α-pinene experiments. Given that OH yield from the decomposition of excited CIs in the isoprene mechanism is lower than that for the α-pinene mechanism, the absence of a detectable interference in the isoprene experiments described here may suggest that the decomposition of these intermediates inside the FAGE detection cell may also be slower. As a result, the observed interference for each alkene is likely proportional to the OH yield from the ozonolysis mechanism. This is consistent with the observation that the observed interference appears to be a constant fraction of the total OH yield in these experiments independent of the ozone concentration and the turnover rate (Figs. S5 and S6). This has been clarified in Section 3.3 of the revised manuscript.

*Pg 10, lines 27- 30: It is interesting/perplexing that the artefact signal is actually lower when the medium inlet is used than when the shorter inlet is used. In light of previous results (Fuchs et al. 2016) which demonstrated a dependence of the magnitude of the artefact signal on cell residence time a comment on the lack of trend in level of interference and inlet length is needed here.*

As pointed out by the reviewer, the level of the observed interference is similar for the short and medium length inlet, but increases with the longest inlet. This suggests that the level of interference is not directly related to the residence time inside the FAGE detection cell, but may be the result of increased collisions with the interior surfaces of the detection cell that occurs when using the longest inlet. We have clarified this here, as well as in the previous discussion in Section 3.1.

*Section 3.3: The working hypothesis on the identity of the observed OH interference is that it derives from the decomposition of a SCI. Were any experiments conducted with alkene concentration in excess? Under these conditions the concentration of the SCI would be maximised, whilst the concentration of externally generated OH from ozonolysis would be small, meaning that the artefact signal should be readily distinguishable from a real OH signal?*

Unfortunately, due to the low vapor pressures of these compounds we were unable to generate high enough concentrations of the alkenes to conduct experiments with them in excess.

*Pg 12, lines 1 – 17: What was the concentration of acetic acid added to the flow-tube? Would any loss of OHss by reaction with acetic acid be expected given the residence time?*

For these experiments, approximately $9 \times 10^{12}$ molecules cm$^3$ of acetic acid was introduced into the flow tube and allowed to react for approximately 200 ms. At this concentration, the reaction with acetic acid was modeled to have a minimal impact on the steady-state concentration of OH, as the rate constant for the OH reaction is approximately $7 \times 10^{-13}$ cm$^3$ molecule$^{-1}$ s$^{-1}$. This has been clarified in the revised manuscript.

*Pg 12, lines 22 – 24: 'Based on these results, the observed OH interference in these experiments could be explained if approximately 5% of these intermediates dissociated..'Does this then effectively disprove the hypothesis that the transmission efficiency of SCI vs OH through the pinhole is substantially different? A comment on transmission efficiency assumed for these lab results vs transmission efficiency estimated from field results (and the implications of these differences) would be welcomed in the revised manuscript.*

It's not clear whether the results of these experiments and modeling disproves the hypothesis that the transmission efficiency of SCIs are substantially different than that for OH, as it is possible that the transmission efficiency of SCIs through the inlet is high, but only 5% of those entering the detection cell actually dissociate into OH.

In these experiments, as well as in field studies, we are assuming that the transmission efficiency of SCIs is essentially 100% and similar to the transmission efficiency of OH. This is based on previous measurements of the loss of OH on different inlet designs and coatings on a similar LIF instrument inlet, as well as measurements of the calibration factor with and without the inlet, suggest that heterogeneous loss of OH on the inlet is minimal (Stevens et al., 1994). This has been clarified in Section 3.3 of the revised manuscript.

Added as suggested

*Pg 4, line 15: add ' with sliding injector' after 'flow tube' so the later discussion on the injector is easier to follow.*

Added as suggested.

Defined as suggested.

*Pg 6, line 20: add ' compared to chemical modulation' after 'spectral modulation'?*

Added as suggested

Changed as suggested.

*Pg 6, line 25: is this 3 – 5 sccm of 1% C3F6 in N2 or 3 – 5 sccm pure C3F6?*

This has been clarified as 3-5 sccm of 99.5% $C_3F_6$.

*Pg 9, line 7: ±0.9 is a very large error. Is this correct?*

The actual error is ±0.09 and has been corrected.

*Pg 10, line 19: Define 'turnover time'*

We have defined the turnover time as the steady-state rate of OH radical propagation, expressed as the alkene ozonolysis rate.

*Figures: Stick to [O3] in molecule cm-3 or ppm.*

We have converted all graphs to concentration units, as suggested.

*Figure 3 – 7: Axes should be rescaled and legends should be made more selfexplanatory. It took me a while to understand what 'Pcell 4' actually represented.*

We have rescaled the graphs and have clarified the legends to make them more self-explanatory, as suggested.

*Figure 3 & 4: It isn't clear to me why the OH yield from the ozonolysis reactions and the OH signal without scavenger are on the same graph? They are two distinct results that just happened to have been determined in the same experiment. I am struggling to suggest a better way to present the results, but maybe the authors could critically review these figures before final publication?*

We included the OH yield from the ozonolysis reactions with the measured OH signal without the $C_3F_6$ scavenger to illustrate the magnitude of the measured interference. We have clarified this reasoning in section 3.1 of the revised manuscript.

*Figure 4: Include a legend on this figure that clearly states the VOC concentration for the different experiments, e.g. green = x cm-3*

We have modified the legend to include the VOC concentrations for each experiment as suggested.

*Figure 5: it is not obvious to me why these three panels are grouped together? The recommendations for improving the figures above should be considered for the figures included in SI also.*

We have grouped the panels in Figure 5 together for simplicity, similar to Figure 6 in Fuchs et al., 2016. We have also changed the figures in the Supporting Information, as suggested.

---

## Author Comment (AC2) · 9 Oct 2017

**Response to Anonymous Referee #2**

We would like to thank the reviewers for their efforts in reviewing this manuscript, and we feel that the manuscript is much stronger with the suggested changes. Below are detailed responses to their comments, which are highlighted in italics.

*This manuscript describes tests performed on the LIF-FAGE instrument in use in the Indiana University to assess the presence of some interference species in the OH radical measurement. The tests performed within this study follow previous work from different LIF-FAGE groups and focus on the ozonolysis of different alkenes. The study shows that an interfering signal is observed during the ozonolysis of specific BVOC but extrapolating these results to ambient concentrations suggest that the interference coming from ozonolysis of unsaturated VOCs will have a negligible impact. I think the topic is of interest as it has been shown how different LIF-FAGE instruments (all with different instrumental parts, flows, etc) react differently to interfering species underlining the need of a characterization for each and every instrument. The manuscript is well written and structured though, in my opinion, it addresses the issue of the interference species insufficiently in-depth and it needs a more extensive characterization and analysis of the results.*

*A first general topic that needs to be address is a characterization of the titration unit used for the investigation of the interfering species. The literature cited when referring to the titration unit does not appear to give a full characterization of the device. As this paper focuses on the interference on the OH radical measurement and as the titration is currently in use in field campaigns, this would be the appropriate study to include the details about the titration unit such as losses on walls, plots with titration efficiency at ambient pressure and low pressure within the detection axis, dependency on the different parameters such as air flow, mixing volume, etc. This could be added in section 2.2.*

We have expanded the description of the chemical titration scheme in this section, including a schematic diagram of the injector ring in Figure 2 as suggested. We have also included a discussion of potential wall losses and the impact of the titration efficiency on the airflow from different inlets.

*The title of the manuscript is misleading. Neither in the abstract or in the conclusion OH radical yields from ozonolysis of selected alkenes are discussed as such. The study focuses mainly on the impact of the interference on the instrument rather than providing new insight in the OH yield. Therefore I feel there should be a more extensive analysis or discussion of possible interfering species. The interference from NO3 as described by Fuchs et al. (2016) is mentioned but this would be the chance to actually perform tests in the laboratory to see how much this particular LIF-FAGE instrument is affected by it. The same is valid for other species as the study from Ren et al. (2004) applies to the LIF-FAGE instrument to which the tests were done. This is a good study to advance the knowledge on the interference species within the OH measurement of the LIF-FAGE by trying new/different possible trace gases.*

We have changed the title of the manuscript as suggested to "Measurements of a potential interference with laser-induced fluorescence measurements of ambient OH from the ozonolysis of biogenic alkenes" as suggested. We hope that this new title will more clearly reflect the overall topic of the paper. Although we agree that additional measurements of potential interferences from other species is needed, we feel that these measurements are beyond the scope of the present paper, which is focused on an interference that may be common to ambient measurements of OH by LIF-FAGE instruments in environments

impacted by biogenic emissions as observed by several groups (Mao et al., 2012; Novelli et al., 2014). Future experiments will involve measurements of potential interference from other species, including $NO_3$ radicals.

*Section 3.3 needs a lot more explanation and clarification: - It is not clear what the hypothesis of the authors is. The first paragraph of this section distinguishes between excited and stabilized Criegee intermediates mentioning that the first produce OH at short times (how short?) and the second at longer time (how long?), Are the authors arguing that the OH radicals they observed in the flow tube are only coming from the excited Criegee intermediates? If that is the case it should be stated explicitly. Though, it would be hard to explain how the OH would be formed within the instrument if the stabilized Criegee intermediates would not decompose within the flow tube. This assumption needs to be check carefully as several studies conclude that the unimolecular decomposition rate of stabilized Criegee intermediates is rather fast (Smith et al., 2016;Chhantyal-Pun et al., 2016).*

We have attempted to clarify the two hypotheses for the source of the interference by separating section 3.3 into two subsections in the revised manuscript. The first subsection (3.3.1 Decomposition of Criegee intermediates produced inside the FAGE detection cell as a source of the interference) attempts to determine whether the source of the interference is due to the production of Criegee intermediates from ozonolysis inside the FAGE detection cell. We have clarified the approximate reaction times associated with OH production from excited Criegee intermediates and stabilized Criegee intermediates as suggested based on the measurements by Kroll et al. (2001), which could explain the observed dependence of the interference on the inlet length and reaction time in the detection cell. However, as discussed in the manuscript, it is unlikely that the interference is due to internal production of Criegee intermediates due to the short reaction time and reduced concentrations of ozone and the alkene.

The second subsection (3.3.2 Decomposition of Criegee intermediates produced outside of the FAGE detection cell as a source of the interference) discusses the possibility that the interference is due to stabilized Criegee intermediates produced from ozonolysis in the flow tube that enter the low pressure FAGE detection cell and decompose into OH radicals. As mentioned in the manuscript, previous measurements have demonstrated that stabilized Criegee intermediates can decompose inside the low pressure region of the FAGE detection cell leading to the formation of OH radicals (Novelli et al., 2014). The experiments involving the addition of acetic acid to the flow tube suggest that the interference is due to stabilized Criegee intermediates produced external to the FAGE detection axis.

As pointed out by the reviewer, the results of these experiments suggest that the majority of the OH radical concentrations observed in the flow tube experiments is due to the rapid decomposition of excited Criegee intermediates, and this has been clarified in the revised manuscript. If all of the OH produced from the flow tube came from the decomposition of stabilized Criegee intermediates, then the OH signal measured in the presence of acetic acid would be less than the measured OH after the interference was subtracted. However, these results do not exclude the possibility that SCIs are also thermally decomposing and contributing to OH production in the flow tube. As pointed out by the reviewer, a decomposition rate of  SCIs that is similar to that measured by Smith et al. (2016) and Chhantyal-Pun et al. (2017) could also compete with reaction of this intermediate with acetic acid. Additional experiments beyond the scope of this paper are needed to determine whether stabilized Criegee intermediates contribute to OH radical production in the ozonolysis mechanisms of these biogenic compounds. This has been clarified in the revised manuscript. However, the absence of OH production from stabilized Criegee intermediates in the atmospheric pressure flow tube does not preclude OH production from these intermediates inside the low pressure FAGE detection cell. The increased collisions with the walls of the

cell as well as the supersonic shock that occurs as the air stream expands into the low pressure region may provide the conditions necessary for dissociation of these intermediates into OH radicals.

*The comparison of the signals as shown in Figure 7 is only valid if the OH formed within the flow tube originates from excited Criegee intermediates only. Formulas (for example, TOT signal = OHFlowTube (OHExcitedCriegee + OHStabilizedCriege +OHInterference) explaining clearly the expected component of every signal should be added to avoid confusion. If the OH observed in the flow tube also originates from stabilized Criegee intermediates it would not be possible to compare full red and full green symbols as the injection of acetic acid would remove a source, within the flow tube, for the OH observed.*

We agree that the results of the experiments suggest that the majority of the OH formed within the flow tube originates from excited Criegee intermediates, and as discussed above this has been clarified in the revised manuscript. Because these experiments cannot distinguish between OH produced in the flow tube from excited or stabilized Criegee intermediates, we have chosen not to separate these components in a formula for simplicity.

*The comparison with the MCM 3.2 needs a lot more detail. The MCM mechanism as is does not include the chemistry needed to do a proper comparison, e.g. which unimolecular rate coefficient was used for the decomposition of the stabilized Criegee intermediate? On which assumptions/studies is the rate coefficient based? How do the authors deal with the fact that one of the two excited Criegee intermediates in the MCM does not decompose forming a stabilized Criegee intermediates (APINAOO)? How is the speciation of the 4 SCI formed from a-pinene treated (e.g. syn vs anti chemistry, relative yields, different unimolecular channels)? Are there additional losses included in the model for the SCI?*

We have added additional details describing the α-pinene ozonolysis mechanism in the MCM as suggested. This version of the MCM does not include a mechanism for the formation of OH from the stabilized Criegee intermediate, nor does it distinguish between the syn and anti isomers of the Criegee intermediates. No additional reactions were added to the model. Despite its shortcomings, the MCM was used to provide a rough estimate of the concentration of Criegee intermediates to compare with the experimental measurements for simplicity.

*Figures need to be revised. In particular legends are not easy to understand rendering the message of the figure not very clear. I would recommend publication in AMT once these general points are addressed.*

We have revised the figures both in the main manuscript as well as in the supplementary information as suggested. In particular, we have clarified the legends so that they are easier to understand.

*Specific comments:*

*Title: As suggested above, the title needs to be change as the focus of the study is the study of the LIF-FAGE interference. The yield of OH radicals from the ozonolysis of BVOCs does not seem to be the main topic of the manuscript.*

As mentioned above, we have changed the title to more clearly reflect the overall topic of the paper.

*Page 2, Lines 9 to 16: The OP3 field campaign results (Whalley et al., 2011) needs to be added.*

We have included the results from the OP3 campaign in the introduction as suggested.

*Page 2, Lines 21 to 27: Tests were done on a specific instrument, it needs mentioning.*

We have clarified that these tests were done on the Penn State instrument.

*Page 3, Line 20: The OH radical concentration measured with the LIF-FAGE agrees with the measurements performed with two CIMS instruments.*

We have clarified this as suggested.

*Page 3, Line 28 to 30: Here the text is misleading. Criegee intermediates decompose forming OH at low pressure and ambient pressure. Several experimental studies are now available proving the decomposition path and suggesting a rate (Smith et al., 2017;Kidwell et al., 2016;Fang et al., 2016) plus extensive theoretical material (see tables in Vereecken and Francisco (2012)).*

We clarified in this discussion that Criegee intermediates can decompose at ambient pressure, and have included the references as suggested.

*Page 4, Line 21: Is this reaction time measured or calculated?*

The reaction time was calculated based on measurements of the flow velocity. This has been clarified.

*Page 4, Line 24: The majority of the tests described in this study include data points collected at 3 ozone values (I assume 1, 2 and 3 ppm). Was it not possible to explore a larger range of ozone values that would make the fit more robust? The accuracy of the ozone measurement needs to be added.*

Unfortunately, the ozone generator used in these experiments provided the highest stability at these concentrations and this is why most of the experiments were done at these three concentrations. We have clarified this in the revised manuscript and have included an estimated uncertainty associated with the ozone measurements.

*Page 4, Lines 26 on: Was any measurement done with, for example, a GC instrument to compare the calculated concentration of VOCs with the measured one? How were the losses on walls accounted for? What is the error on the estimated concentration?*

Unfortunately, no direct method for measuring the concentration of these BVOCs was available, and as a result the absolute concentration of BVOCs in the flow tube is highly uncertain due to potential wall losses prior to entering the flow tube. This has been clarified in the revised manuscript.

*What are the concentrations for isoprene and MBO? A table summarizing the different experiments, at which conditions they were performed and the amount of interference observed would be helpful.*

We have included the estimated concentrations for isoprene and MBO in the revised manuscript, and have included a table summarizing the conditions of the different experiments in the Supplementary Information.

*Page 5, Line 12 and 16: It is not clear which laser is the new one as both lasers have the same identification number.*

We clarified which of the two laser systems was used in these experiments.

*Page 5, Line 20: Is there a particular reason to use a 12 m fiber in the laboratory? Do the authors expect a dependency of the interference on the length of the optical fiber?*

We use a 12 m fiber for these experiments as that is the fiber length used in most field measurements by our instrument. Different fiber lengths result in different reflections from the ends of the fiber, which impacts the background signal. We do not expect that the interference would depend on the length of the fiber.

*Page 6, Line 1 to 2: How long does it take for the OH concentration to stabilize?*

We have clarified that it took several minutes for the OH concentration to stabilize, primarily due to stabilization of the generated ozone concentration.

*Page 6, Line 5: Why are the experiments performed in N2 and not synthetic air?*

Nitrogen from liquid boil-off was used instead of air to reduce the concentration of reactive impurities in the system. This has been clarified in the revised manuscript.

*Page 6, Line 13: "the limit of detection was approximately between. . .". Summarizing the different sensitivity of the instrument for different parameters and inlet configuration in a table rather than a plot would be helpful and the error on the values should be stated.*

We have included a table in the Supplementary Material summarizing the sensitivity and limits of detection for the different experimental parameters, as suggested. We have also included a statement in the revised manuscript on the estimated uncertainty associated with the UV-water photolysis calibration method (± 36%, 2σ).

*Page 6, Section 2.2: As underlined above, a more in detailed characterization of the titration unit with figures of the scavenging experiments, wall losses values, dependency of the OH scavenging on the flow of air sampled, and on the mixing volume, etc., need to be added.*

As discussed above, we have expanded the description of the chemical titration scheme, including a schematic diagram of the injector ring in Figure 2 as suggested. We have also included a discussion of potential wall losses and the impact of the titration efficiency on the airflow from different inlets. Experiments conducted under different lengths of the injector above the inlet did not reveal any loss of ambient OH or improvement in the scavenging efficiency, nor was any change in the scavenging efficiency observed on the flow of air sampled for the two inlet diameters. This has also been clarified in the revised manuscript.

*Page 7, Line 10: For which conditions was the steady state reached in 20 ms? The figures need to be self-explanatory. More text needs to be added in the figure caption together with a clearer legend.*

Computer simulations indicated that under the BVOC and ozone concentrations used in these experiments, the OH concentrations reached steady state in less than 20 ms. This has been clarified in the revised manuscript.

*Page 7, Line 16: For consistency: kO3+VOC and kOH+VOC.*

This has been changed as suggested.

*Page 7, Line 18: How much are kwall and kOH+O3x[O3] for the experiments performed in this study?*

We have added the loss rates due to reaction with the wall and with ozone to the revised manuscript as suggested.

*Page 7, Line 18: "measured as describe in previous work by Handen et al., 2014..".*

This has been changed as suggested.

*Page7, Line 23: In figures 3 and 4, why are the data point with and without C3F6 added for a certain pressure in the cell at different ozone values? Those points are taken consequently or? Is the variation in the ozone due to instability of the ozone generator? Is one data point in the plot the average of a single experiment or the average of the repetition of different experiments performed at the same conditions? What kind of fit is applied? Is it weighted on the errors? Does it account for errors on both x and y axis?*

As pointed out by the reviewer, the points with and without $C_3F_6$ addition in these plots appear at different ozone values due to the instability of the ozone generator. Each measurement was repeated several times and the points represent averages of the measurements. During the measurements, variations in the ozone concentration produced by the generator led to the variations in the measured ozone concentration. This has been clarified in the revised manuscript.

The measured OH yields were determined from a weighted fit of the slope of the plot of the OH concentration versus ozone concentration. The weights are determined from the precision of the measurements of both OH and ozone. This has also been clarified in the revised manuscript.

*Page 8, Line 9: The fit showed in the central and bottom panel of figure S3 hardly represents the data. Was here used a different fit? Is there an explanation for the extremely higher values for the interference for the long inlet with the 1 mm pinhole (bottom panel figure S3) compared to the values observed for the same inlet with the 0.6 mm pinhole (bottom panel figure 3)? The values for the other two inlet length did not show a drastic variation between the 2 different pinholes. Could this be related to a smaller drop in sensitivity observed between medium and long inlets with the 1 m pinhole compared to the 0.6 mm one?*

The lines in the plots do not represent the weighted fits of the data, but represent the expected OH concentrations as a function of ozone concentration based on recommended yields for each reaction. This has been clarified in the revised manuscript.

The larger interference observed with the larger 1mm nozzle diameter and the long inlet length is likely due to the longer reaction time inside the detection cell, resulting in greater collisions with the wall of the longer inlet. This has been clarified in the revised manuscript.

*Page 8, Line 14: "The different pressures in these experiments. . ."*

This has been changed as suggested.

*Page 8, Line 15: Remove likely*

This has been removed as suggested.

*Page 8, Lines 18 to 20: The longer inlet will also increase the OH losses so it is still not clear why with the longer inlet there is an increase of interference when increasing the pressure.*

As pointed out by the reviewer, the longer inlet also increases the wall loss of OH radicals in the detection cell. Losses of OH produced outside of the detection cell are accounted for during the calibration of the instrument, but losses of OH produced inside the detection cell are not. As a result, the observed interference likely represents a lower limit to the actual interference produced in the detection cell. One possible explanation of the observation of an increased interference with the longer inlet is that the loss of

OH is less than rate of production of the interference under these conditions. This has been clarified in the revised manuscript.

*Page 8, Lines 20 to 21: A legend should be added to figure 4 together with the errors on the concentrations of the BVOC.*

We have added a legend listing the estimated BVOC concentrations. As discussed above, these estimated concentrations are highly uncertain given potential losses prior to entering the flow tube. This has been clarified in the revised manuscript.

*Page 8, Lines 27 to 28: By looking at figure 4 central panel it is possible to observe a trend with higher concentration of interference for higher concentration of β-pinene.*

We have added a statement indicating that there appears to be a trend in the measured OH yield with increasing β-pinene concentration.

*Page 8, Line 30: Which Ocimene isomer was used during the experiments?*

The ocimene used in these experiments contained a mixture of isomers. This has been clarified.

*Page 9, Line 9: Remove likely.*

This has been removed as suggested.

*Page 9, Line 14: Was the expected steady state concentration of OH radicals for the condition of the experiments calculated? Rate coefficient of Isoprene with O3 is less than a factor of 2 slower than the one with β-pinene and the tabulated OH yields are similar for both (~0.25) so it is not clear why there would not be any detectable OH signal especially at the highest ozone concentration.*

We have clarified that the lower expected steady-state OH concentration in the ozonolysis of isoprene and MBO are lower due to the relative reactivity with ozone and OH as well as the overall OH yield. We have also performed simulations that show that the expected OH concentration in the isoprene experiments were approximately 50 times lower than that for the α-pinene experiments, consistent with a lower steady-state OH concentration estimated using Equation 1, and near or below the detection limit of the instrument. This has also been clarified in the revised manuscript.

*Page 10, Line 10: To which experiments does "These experiments" refer to?*

This has been clarified.

*Page 10, Line 15: Use molecules cm-3 for the x axis as in the previous plots. Here it would be interesting to also have in a figure the amount of interference from β-pinene and ocimene.*

We have changed the units in this figure as suggested, and have added the results from the ocimene experiments to this plot.

*Page 10, Line 18: Ocimene and β-pinene should be added to Figure 6 to see if they lie on the same line as it is shown in Fuchs et al. (2016). The possible explanation about the large scatter observed in the data should also be given. How does the plot look like for longer inlets?*

We have added the results for ocimene and β-pinene to Figure 6 as suggested. The large scatter observed in the data are likely due to the large uncertainty associated with estimates of the BVOC concentrations in these experiments. However, the level of the observed interference is greater than that illustrated in Fig. 6

for the measurements using the long inlet, suggesting the similarity with the results of Fuchs et al. (2016) may be fortuitous, as differences in the design of the instrument impacts the level of the interference. In contrast to the results of Fuchs et al., the level of interference as a function of the turnover rate is not similar for all of the BVOCs tested. While the observed interference as a function of turnover rate appears to be similar for the ozonolysis of $\alpha$-pinene and ocimene, the observed interference for the ozonolysis of $\beta$-pinene is significantly less. This may also be related to uncertainties associated with estimates of the concentration of $\beta$-pinene in the flow tube, but may also suggest differences in the mechanism for the production of the interference for some BVOCs. Additional measurements of the interference for other BVOCs are needed to resolve this discrepancy. These issues have been clarified in the revised manuscript.

*Page 10, Line 24 on: Here a table including also β-pinene (or is there a reason not to list it?) could substitute the plots.*

We have included a table summarizing the measurements in Figure S5 in the Supplementary Material as suggested.

*Page 11, Line 3 on: The last paragraph is difficult to understand. I see what point the authors want to make (although I am not sure this is the appropriate place in the manuscript to make this point) but the text could benefit from rephrasing.*

We have rephrased this paragraph as suggested.

*Page 11, Lines 11 to 15. Criegee intermediates can also be formed directly in a stabilized form from non-endocyclic double bonds. It would also be helpful at this stage to give an estimate of the time scale where these CI give OH, i.e. stabilized CI of the order of milliseconds, collisional stabilisation is of the order of 108 s-1, prompt decomposition is thus at even faster rates, also implying a very low steady state concentration of excited CI. The OH concentrations are not in steady state because the excited CI are in steady state: these two species have different formation and destruction timescales, where excited CI will reach steady state concentration orders of magnitude faster than OH. It also needs to be specified that the steady state is reached only within the flow tube and not inside the instrument. In the conclusions the authors also suggest that SCI decomposition may not be constant throughout the detection cell, e.g. more SCI migrating to the walls and only then undergoing decomposition. Such effects should also be discussed in more detail at the start so a complete kinetic model is available prior to interpreting the results.*

We have added information regarding the estimated rates of decomposition of both excited Criegee intermediates and stabilized Criegee intermediates to this section as suggested, and have specified that the OH radical reaches steady-state in the flow tube rather than inside the detection cell in the following paragraph, as suggested. We have also included a statement clarifying that the lifetime of SCIs inside the low pressure detection cell may be long enough for wall collisions, as suggested.

*Page 12, Line 5: Was it not possible to try different SCI scavengers like water and/or SO2?*

Acetic acid was chosen over $SO_2$ as an SCI scavenger as $SO_2$ fluorescence at 308 nm can interfere with OH measurements (Fuchs et al., 2016). Addition of water in the presence of the high concentrations of ozone in these experiments would have led to significant laser-generated OH from the photolysis of ozone followed by reaction of the $O(^1D)$ product with water vapor. This has been clarified in the revised manuscript.

*Page 12, line 24 to 26. The modeling of the concentration of excited CI should go to the section at the end of page 11 to further strengthen the assertion that excited CI cannot be the source of the interference. This should also be stated explicitly.*

The modeled concentration of the excited CI in this section refers to the steady-state concentration achieved under the concentrations of ozone and the alkene in the flow tube, which are much greater than the concentration inside the detection axis. Because the discussion at the end of page 11 in the original manuscript addresses the production of excited CI inside the detection axis, we have not moved this discussion. We have stated explicitly in both sections that excited CI cannot be the source of the interference as suggested.

*Page 12, Lines 28 "Criegee..".*

This typo has been corrected.

*Page 13, Lines 2 to 3: Chao et. al 2015 measured a fast rate coefficient for CH2OO with water dimers but it is not a good idea to generalize this rate for all the Criegee intermediates as several studies (see Vereecken et al. (2015) and citations therein) have shown that the rate with water and water dimers will strongly depend on the structure of the Criegee intermediates.*

We have clarified this in the revised manuscript, as suggested.

*Page 13, Line 6: The authors discuss the potential impact of using alkene ozonolysis on FAGE calibration. It could be beneficial to separate that out in a separate paragraph or even a separate section.*

As suggested, we have created a separate section in the revised manuscript discussing the potential impact of interferences associated with using the alkene ozonolysis technique to calibrate LIF-FAGE instruments.

*Page 14, Line 14: The Isoprene concentration is mentioned as indicative on the likelihood of interferences in this specific LIF-FAGE instrument but as no interference was observed for concentrations way larger than what observed in the field it does not seem to be the appropriate parameter.*

We have clarified this discussion to focus on how the difference in temperature between the different field campaigns likely led to different BVOC concentrations, using isoprene as an indicator of this difference.

*Page 14, Line 23: What was the percentage of the interference compared to the "real" atmospheric OH? How much was the known ozone interference?*

During this campaign, all interferences accounted for approximately 60% of the total measured OH signal on average. The known interference from laser-generated OH varied with laser power, ambient ozone and water concentrations, but was approximately half of the total measured interference on some days, while on other days accounted for all of the measured interference. These results will be summarized in a future publication, but we have added this information to the revised manuscript as suggested.

*Page 14, Lines 29 to 31: Any hypothesis on what could be the cause for the interference?*

Similar to that discussed in Novelli et al. (2016), the results of these measurements does not allow for an unequivocal identification of the source of the interference, although it may be possible that the interference observed during ambient measurements may be due to the ozonolysis products of BVOCs not

tested in the experiments reported here. This hypothesis has been added to the revised manuscript as suggested.

*Page 15, Lines 8 to 10: This sentence is correct if contemporary to the addition of Acetic acid also C3F6 was injected and no OH signal was observed. Was this the case?*

Unfortunately we did not conduct tests where both acetic acid and $C_3F_6$ were added simultaneously. However, the observation that the measured OH concentrations after addition of acetic acid were similar to measurements when the interference was subtracted and suggests that the interference in these experiments is due to stabilized Criegee intermediates. This sentence has been revised to clarify this point.

*Page 15, Lines 18 to 20: Trying to minimize any interference is important, but isn't this specific interference actually, probably, not so relevant in field campaigns? Is there going to be a gain in modifying the instrument for a negligible interference risking losing in sensitivity or encountering different problems? Are there tests showing that the modification improve the quality of the OH measurement?*

Based on estimates of stabilized Criegee concentrations in the ambient atmosphere, it appears that the interference characterized in these experiments may not be relevant in field campaigns. However, it is possible that other unknown interferences may have a similar dependence on instrument parameters. We have revised this statement to reflect that tests will be done to minimize the observed ambient interference though changes in both reaction time and potential surface collisions, while also maintaining the quality of the ambient OH measurement.